# Learning Continuous and Discrete Dynamics for Time Series Anomaly Detection via Probabilistic Modeling

## Abstract

Anomaly detection for multivariate time series plays an important role in many applications, enabling, e.g., risk monitoring in cyber-physical systems. While existing methods achieve good results on continuous variates, they struggle when having to learn both continuous and discrete dynamics across continuous time. Further, existing methods simply sum up reconstruction or contrastive errors from each variate to obtain final anomaly scores without recognizing differences in importance of variates with different measurement units. To overcome these limitations, we propose TAD-UP that learns both continuous and discrete dynamics for Time series Anomaly Detection via Unified Probabilistic modeling. First, we propose two co-dependent branches of efficient neural ordinary differential equations with the compound Poisson process to learn both continuous and discrete dynamics for different variates. We also propose a gate mechanism to learn correlations among different dynamics. Second, we propose to model a joint probability distribution for anomaly detection. The resulting model is optimized using Maximum Likelihood Estimation on joint variates, instead of using reconstruction or contrastive losses on each variate. We detect anomalies using joint probabilities, which take the marginal probabilities of different variates into account. Experiments on nine real-world datasets from different domains offer evidence that TAD-UP is capable of state-of-the-art accuracy and better efficiency tradeoff.

## 1 Introduction

Cyber-physical systems employ sensors to monitor their environment, thereby producing multi-variate time series data, consisting of timestamped sequences of vectors that encompass multiple variates with different physical and mathematical meanings (Cirstea et al., 2019; Campos et al., 2023; Liu et al., 2024; Zhang et al., 2023). For example, computing systems may record a variety of runtime indicators (Abdulaal et al., 2021; Su et al., 2019), and environmental systems may record weather and water information (Lai et al., 2021; Ghorbani et al., 2024). Detection of anomalies in multivariate time series is essential in many real-world applications, as it supports fault analyses and facilitates risk monitoring to ensure the normal operation of cyber-physical systems, thereby avoiding economic losses (Jhin et al., 2023; Cerqueira et al., 2023; Hojjati et al., 2023).

The multiple variates in time series can generally be categorized into continuous and discrete variates, as shown in Figure 1. Continuous variates are represented by real numbers and include, *e.g.,* service latencies, memory and CPU usages. Their values are constrained with continuity, as illustrated in Figure 1(a). They thus exhibit *continuous dynamics*. Discrete variates are represented by natural numbers and are not amenable to mathematical calculations; thus, adding or subtracting numerical discrete values are not meaningful. Discrete variates can encode categories, flags, and status information of dynamic systems. Their values can jump and are constrained with discreteness, as illustrated in Figure 1(b). They thus exhibit *discrete dynamics*. In addition, different variates are recorded using different measurement units. For example, memory usages are measured in GB, CPU usage rates are measured in percentages, temperatures are measured in degrees Celsius, and category information is encoded in natural numbers.

Classic (Breunig et al., 2000; Hariri et al., 2019; Liu et al., 2024; Zhang et al., 2023) and deep learning-based (Ruff et al., 2018; Fang et al., 2024; Campos et al., 2022; Zong et al., 2018) anomaly

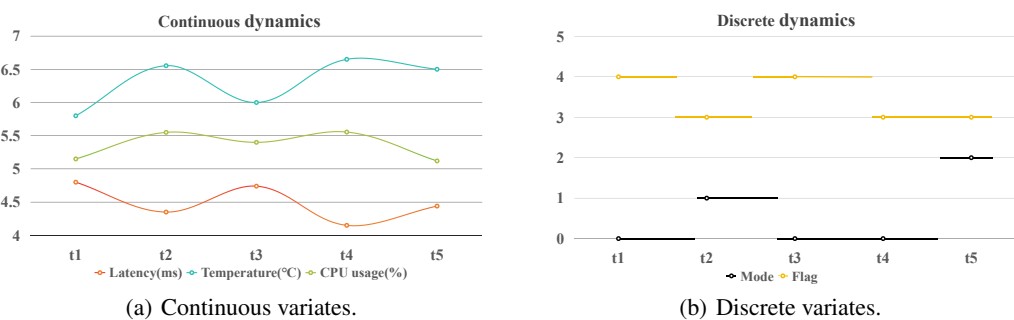

(a) Continuous variates.      (b) Discrete variates.

Figure 1: Continuous and discrete variates.

detection methods learn normal patterns from unlabeled data and achieve good results when applied to regular time series with continuous variates. However, all existing methods (Schmidl et al., 2024; Zhao et al., 2022) remain unable to discriminate and learn continuous and discrete dynamics to improve multivariate time series anomaly detection (He et al., 2023) when breaking the continuity or discreteness property can indicate more likely abnormal. In addition, existing methods (Zong et al., 2018; Jhin et al., 2023; Schmidl et al., 2024; Zhao et al., 2022; Fang et al., 2024; Ghorbani et al., 2024; Shentu et al., 2025) are unable to determine the importance of each variate when having to combine information across variates as part of identifying anomalies at a timestamp. They simply sum up per-variate errors to obtain anomaly scores and determine anomalies. The above problem is characterized by two challenges:

**Challenge 1:** Continuous and discrete dynamic variates behave very differently over time. This also prevents existing methods from learning correlations across such different variates. Although PAD (Jhin et al., 2023) utilizes neural controlled differential equations (NCDEs) (Kidger et al., 2020) with interpolation methods to learn continuous dynamics for irregular time series anomaly detection, PAD cannot learn discrete dynamics where variates can jump. PAD also cannot model correlations across different dynamics, although variates with such different dynamics may be correlated as shown in Figure 1 where variates latency and flag are correlated. In addition, PAD must map original time series with discrete timestamps into controlled paths using interpolation methods, which introduces additional processing steps and complexity. As a result, it is challenging to learn continuous and discrete dynamics and correlations effectively and efficiently.

**Challenge 2:** Existing methods cannot learn the relative importance of different variates on final anomaly scores without supervision signals. Anomaly detection must often operate unsupervised because labeled data is unavailable in real-world settings, where manual labeling is infeasible (Schmidl et al., 2024; Ghorbani et al., 2024; Liu et al., 2024). Thus, no supervision signals are available to existing methods to learn importance. The reconstruction or contrastive errors from different variates belong to semantic spaces with different measurements, and summing up all errors ignores differences in importance across variates.

To contend with these challenges, we propose **TAD-UP** for learning interactive continuous and discrete dynamics for multivariate **T**ime series **A**nomaly **D**etection via **U**nified **P**robabilistic modeling, by employing novel neural co-dependent ordinary differential equations (co-ODEs). To address **Challenge 1**, we propose two co-dependent branches of neural ordinary differential equations (NODEs) with compound Poisson process to learn correlated continuous and discrete dynamics across different variates together. Further, we propose gate temporal convolution networks (TCNs) to model correlations among discrete and continuous variates. Instead of NCDEs requiring complex interpolation methods and controlled paths, our method learns continuous and discrete dynamics and their correlations effectively and efficiently . To address **Challenge 2**, we model the joint probability distribution and obtain final anomaly scores from a unified probabilistic space. Specifically, we use the multivariate Gaussian distribution to model the observation probabilities of continuous variates, and we use the softmax distribution to model the observation probabilities of discrete variates. We optimize the model with Maximum Likelihood Estimation (MLE) in the unified probabilistic space. Finally, we detect anomalies using the joint probabilities that take the marginal probabilities of different variates into account. A low joint probability of all observations at a timestamp indicates a

likely anomaly. A low marginal probability of one variate at a timestamp indicates that this variate is important for the detected anomaly. We summarize the contributions as follows.

- We present the first method capable of learning both continuous and discrete dynamics for multivariate time series anomaly detection.
- We propose a novel design encompassing gated co-dependent NODEs with compound Poisson process to learn correlated continuous and discrete dynamics.
- We model joint probability distribution across different dynamics and optimize the model with MLE, which helps obtain unified anomaly scores that consider the importance of different variates.
- Experiments on nine datasets from different domains offer evidence that TAD-UP is capable of state-of-the-art accuracy and better efficiency tradeoff, even compared with the large pre-training model DADA.

## 2 RELATED WORK

### 2.1 TIME SERIES ANOMALY DETECTION

Recent methods focus mainly on unsupervised detection based on one-class classification theory (Zhao et al., 2022; Ruff et al., 2018). The clustering-based methods, such as K-means (Schmidl et al., 2024; Liu et al., 2024; Zhao et al., 2022), PCA (Liu et al., 2024), OCSVM (Ruff et al., 2018), and THOC (Shen et al., 2020), aim to cluster normal samples around a center. The density estimation-based methods, such as LOF (Breunig et al., 2000; Zhang et al., 2023), IForest (Hariri et al., 2019), DAGMM (Zong et al., 2018), LODA (Pevný, 2016), and HBOS (Goldstein & Dengel, 2012), assume that abnormal samples lie in a low-density region. LSTM (Schmidl et al., 2024) proposes forecasting-based approaches that determine whether there are anomalies based on forecasting errors. The Autoencoder (Hao et al., 2024; Fang et al., 2024) is a classic architecture used in reconstruction-based methods, such as AE (Fang et al., 2024), GPT4TS (Zhou et al., 2023), MEMTO (Song et al., 2024), ModernTCN (donghao & wang xue, 2024), and SensitiveHUE (Feng et al., 2024). Omni (Su et al., 2019) and InterFusion (Li et al., 2021) utilize variational networks. BeatGAN (Du et al., 2023) introduces a discriminator to help generate and reconstruct normal data. CAE-Ensemble (Campos et al., 2022) proposes to leverage diverse AEs for reconstruction. DADA (Shentu et al., 2025) pre-trains a foundation model with large data using TCN-based autoencoders. ImDiffusion (Chen et al., 2023) and D3R (Wang et al., 2023) utilize diffusion-based networks to generate normal data. Contrastive learning-based methods DCdetector (Yang et al., 2023) and AnomalyTransformer (Xu et al., 2022) learn similar hidden features for normal samples. However, none of these methods discriminate between continuous and discrete variates, and they cannot learn continuous and discrete dynamics for different variates.

### 2.2 MULTI-VARIATE STRATEGIES

The Channel-Independent (CI) strategy ignores correlations among variates and encompasses methods such as DCdetector (Yang et al., 2023) and DADA (Shentu et al., 2025). These methods use the same model on different variates in parallel. The Channel-Dependent (CD) strategy models correlations among different variates and encompasses methods such as ModernTCN (donghao & wang xue, 2024) and CAE-Ensemble (Campos et al., 2022). These methods use attention mechanisms or graph neural networks to capture correlations. However, none of these methods are able to learn the importance of different variates for detecting anomalies without supervision signals.

### 2.3 NEURAL DIFFERENTIAL EQUATIONS

Chen et al. (2018) were the first to propose to learn continuous dynamics along continuous time with NODEs. These have since been employed for time series forecasting, including for traffic (Jin et al., 2023) and climate (Jia & Benson, 2019) forecasting. The recently proposed NCDEs utilize the Riemann-Stieltjes integral (Kidger et al., 2020) to control the continuous dynamics using interpolation path from the time series. PAD (Jhin et al., 2023) utilizes NCDEs and interpolation methods to learn continuous dynamics for irregular time series, and uses contrastive learning with data augmentation to detect anomalies. However, none of these proposal are able to learn continuous and discrete dynamics and correlations among variates.

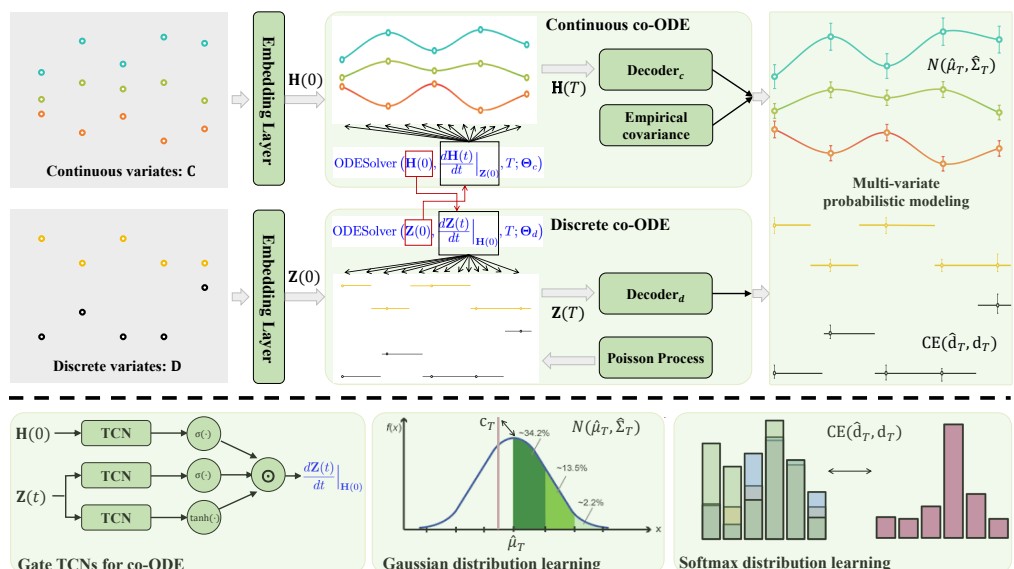

Figure 2: The main architecture and modules.

## 3 PRELIMINARIES

We formalize the problem. We provide background on neural differential equations in Appendix A. Frequently used notation is summarized in Appendix Table 4.

**Multi-variate Time Series** is represented as $(\mathbf{C}, \mathbf{D})$, where $\mathbf{C} = \langle \mathbf{c}_1, \mathbf{c}_2, \cdots, \mathbf{c}_T \rangle \in \mathbb{R}^{N_C \times T}$ records observations of $N_C$ continuous variates using real numbers for each of $T$ timestamps and $\mathbf{D} = \langle \mathbf{d}_1, \mathbf{d}_2, \cdots, \mathbf{d}_T \rangle \in \mathbb{N}^{N_D \times T}$ records observations of $N_D$ discrete variates using natural numbers for each of $T$ timestamps. Thus, $\mathbf{c}_i \in \mathbb{R}^{N_C}$ and $\mathbf{d}_i \in \mathbb{N}^{N_D}$ are continuous and discrete observations, respectively, at timestamp $t_i$, $i \in \{1, 2, \cdots, T\}$. In the special case where $N_D = 0$, $\mathbf{C}$ is the kind of time series considered in previous studies that consider only continuous variates with observation from $\mathbb{R}$ (Jhin et al., 2023; Campos et al., 2022; Shentu et al., 2025). We use $\mathbf{C}^j \in \mathbb{R}^T$, $\mathbf{c}_i^j \in \mathbb{R}^1$, $\mathbf{D}^j \in \mathbb{N}^T$ and $\mathbf{d}_i^j \in \mathbb{N}^1$ to indicate observations of the $j$-th variate.

**Time Series Anomaly Detection.** Anomaly detection identifies the timestamps in a time series that contain anomalous observations, *i.e.,* a binary classification of all timestamps is produced. With sliding windows (Shentu et al., 2025), anomaly detection takes as inputs time series $(\mathbf{C}, \mathbf{D})$, and outputs a binary vector $\mathbf{Y} = \langle y_1, y_2, \cdots, y_T \rangle$ where $y_i = 1$ indicates an anomaly at timestamp $t_i$.

## 4 METHODOLOGY

### 4.1 OVERALL ARCHITECTURE

We present the main architecture and modules of TAD-UP in Figure 2. **The embedding layers** take as inputs time series data $(\mathbf{C}, \mathbf{D})$, and maps all continuous observations and discrete observations at each timestamp $t_i$ to $d$-dimensional vectors, to extract dense features for each discrete timestamp. Then, $\mathbf{H}(0)$ is the hidden state feature of the initial condition for the continuous co-ODE, and $\mathbf{Z}(0)$ is the hidden state feature of the initial condition for the discrete co-ODE. **The continuous co-ODE** takes the continuous variate feature $\mathbf{H}(0)$ as the initial condition, uses gated TCNs on the discrete variate feature $\mathbf{Z}(0)$ to learn the correlations of how the discrete variates influence the continuous variates, and outputs final hidden state feature $\mathbf{H}(T)$ that captures correlated continuous dynamics for continuous variates. **The discrete co-ODE** takes the discrete variate feature $\mathbf{Z}(0)$ as the initial condition, uses gated TCNs on the continuous variate feature $\mathbf{H}(0)$ to learn the correlations, and uses the compound Poisson process to output final hidden state feature $\mathbf{Z}(T)$ that captures correlated discrete dynamics that can jump for discrete variates. **The multi-variate probabilistic modeling** outputs the estimated joint probability of all observations at final timestamp $T$. The decoders take as inputs $\mathbf{H}(T)$ and $\mathbf{Z}(T)$, output an estimated Gaussian mixture distribution $N(\hat{\mu}_T, \hat{\Sigma}_T)$ for the observations of the continuous variates, and output an estimated softmax distributions $\hat{\mathbf{d}}_T$ for the

observations of discrete variates. Finally, the model are trained using MLE in the unified probabilistic space, and low joint probabilities of all observations at a timestamp indicate a likely anomaly.

## 4.2 THE EMBEDDING LAYERS

The embedding layers take as inputs time series $(\mathbf{C}, \mathbf{D})$. We first apply standard normalization (Ulyanov et al., 2016) on the observations of each continuous variate to learn stable features (Kim et al., 2021), *i.e.,* each $\mathbf{C}^j \in \mathbb{R}^T$ being standard normalized. Then, the embedding layers map the normalized observations $\mathbf{c}_i \in \mathbb{R}^{N_C}$ and the observations $\mathbf{d}_i \in \mathbb{R}^{N_D}$ to $d$-dimensional vectors $\mathbf{h}_i \in \mathbb{R}^d$ and $\mathbf{z}_i \in \mathbb{R}^d$, to extract the dense features at each discrete timestamp $t_i$ for continuous and discrete variates, respectively.

The embedding layer for continuous variates is a fully connected feed-forward layer (FC):

$$\mathbf{h}_i = \text{FC}(\mathbf{c}_i), \text{ for } i \in \{1, 2, \cdots, T\}, \ \mathbf{H}(0) = (\mathbf{h}_1, \mathbf{h}_2, \cdots, \mathbf{h}_T), \tag{1}$$

where $\mathbf{H}(0) \in \mathbb{R}^{d \times T}$ denotes the hidden feature for the initial condition of the continuous co-ODE.

The embedding layer for discrete variates is one-hot based lookup embedding layer (implemented by Pytorch EmbeddingBag), where discrete variates are presented using natural number encoding:

$$\begin{aligned} \mathbf{d}^j &\leftarrow \text{one-hot}(\mathbf{d}^j), \text{ for } 1 \le j \le N_D, \\ \mathbf{z}_i &= \text{lookup-embedding}(\mathbf{d}_i), \text{ for } i \in \{1, 2, \cdots, T\}, \ \mathbf{Z}(0) = (\mathbf{z}_1, \mathbf{z}_2, \cdots, \mathbf{z}_T), \end{aligned} \tag{2}$$

where $\mathbf{Z}(0) \in \mathbb{R}^{d \times T}$ denotes the hidden state feature for the initial condition of the discrete co-ODE.

## 4.3 THE CONTINUOUS CO-ODE

The continuous co-ODE aims to learn fine-grained and continuous temporal dynamics. Given the shortcomings of RNNs, including high time-complexity and gradient explosion (Jin et al., 2023), we choose temporal convolution networks as the function $f(\cdot; \boldsymbol{\Theta})$ to estimate the continuous dynamic differentiation of $\mathbf{H}(t)$ along the continuous time $t$:

$$\mathbf{H}(t) = \mathbf{H}(0) + \int_0^t \frac{d\mathbf{H}(s)}{ds} ds, \ \frac{d\mathbf{H}(t)}{dt} = \tanh\Big(\text{TCN}\big(\mathbf{H}(t); r, k\big)\Big), \tag{3}$$

where $\text{TCN}(\cdot; r, k)$ is a Conv1d temporal convolution layer that is performed along the last dimension of $\mathbf{H}(t)$, *i.e.,* the discrete timestamp dimension. The channel of the TCN layer is set to be $d$, equal to the dimensionality of feature vector, and $\tanh(\cdot)$ is the hyperbolic tangent activation function. Finally, $r$ and $k$ are the dilation factor and the convolution kernel, respectively. To maintain the same dimensions for the continuous hidden features $\mathbf{H}(t)$, we use the padding-based Conv1d temporal convolution layer. We omit $r$ and $k$ in the following.

In order to learn the influence among different variates, we propose a gated TCN architecture as the attention mechanism to model the conditional information:

$$\frac{d\mathbf{H}(t)}{dt}\Big|_{\mathbf{Z}(0)} = \tanh\Big(\text{TCN}_1\big(\mathbf{H}(t)\big)\Big) \odot \sigma\Big(\text{TCN}_{\text{gate}_1}\big(\mathbf{H}(t)\big)\Big) \odot \sigma\Big(\text{TCN}_{\text{gate}_2}\big(\mathbf{Z}(0)\big)\Big), \tag{4}$$

where $\odot$ denotes the element-wise Hadamard product, $\text{TCN}_{\text{gate}_1}$ and $\text{TCN}_{\text{gate}_2}$ are the gate convolutions used for the continuous variates and discrete variates, respectively, and $\sigma(\cdot)$ is the sigmoid activation function that is used control the amount of information flows among different variates.

Then, we can obtain the feature $\mathbf{H}(T) = \text{ODESolver}\big(\mathbf{H}(0), \frac{d\mathbf{H}(t)}{dt}\big|_{\mathbf{Z}(0)}, T; \boldsymbol{\Theta}_c\big)$ (Jia & Benson, 2019) that learns conditional continuous dynamics, where $T$ is the time length of the last timestamp and $\boldsymbol{\Theta}_c$ denotes the parameters in the continuous co-ODE branch.

## 4.4 THE DISCRETE CO-ODE

We propose a compound Poisson process to learn the discrete dynamics. First, a Poisson process is a time point generative model that outputs a sequence of discrete eventual time points $\mathcal{H} = \{\tau_{t_i}\}$. The $\tau_{t_i}$ denotes that there is a discrete eventual time point at time $t_i$ and no eventual time points after time $t_{i-1}$ and before time $t_i$. Thus, $\tau_{t_i}$ can indicate whether there is a jump at time $t_i$. A Poisson

process can be defined formally by a counting function $\mathbf{N}(t)$ along continuous time, as exemplified in Appendix Figure 3. $\mathbf{N}(t)$ records the number of eventual time points before time $t$, and is defined as follows:

$$\mathbf{N}(t) = \sum_{\tau_{t_i} \in \mathcal{H}} He(t - t_i), \text{ where } He(t) = \begin{cases} 0 & t \leq 0 \\ 1 & \text{otherwise,} \end{cases} \tag{5}$$

where $He(\cdot)$ is the Heaviside step function. The probability of the next eventual time point after time $t$ usually depends on the contexts before time $t$, known as the rate or intensity of the Poisson process. Such temporal dependencies can be described by a conditional intensity function $\lambda(t)$. The larger $\lambda(t)$ is, the sooner the next eventual time point will occur, as exemplified in Appendix Figure 3, where a larger $\lambda$ results in more possible and sooner eventual time point. Formally, let $\mathcal{H}(t)$ denote the previous contexts of eventual time points up to, but not including, time $t$. Then $\lambda(t)$ defines the probability of observing the next eventual time point conditioned on the history, and the probability distribution is formally defined by:

$$\mathbb{P}\left\{ \text{eventual time point in } [t, t + dt) \mid \mathcal{H}(t) \right\} = \lambda(t) \cdot dt \tag{6}$$

With this Poisson process, we propose the following compound discrete co-ODE:

$$\lambda(t) = \text{FC}_\lambda\left(\mathbf{Z}(t)\right), \ \mathbf{Z}(t) = \mathbf{Z}(0) + \int_0^t \frac{d\mathbf{Z}(s)}{ds} d\mathbf{N}(s),$$

$$\frac{d\mathbf{Z}(t)}{dt}\Big|_{\mathbf{H}(0)} = \tanh\left(\text{TCN}_2\left(\mathbf{Z}(t)\right)\right) \odot \sigma\left(\text{TCN}_{\text{gate}_3}\left(\mathbf{Z}(t)\right)\right) \odot \sigma\left(\text{TCN}_{\text{gate}_4}\left(\mathbf{H}(0)\right)\right), \tag{7}$$

where $\text{FC}_\lambda$ denotes a fully connected feed-forward layer that learns $\lambda(t)$ from the contexts $\mathbf{Z}(t)$ to control the conditional intensity of the next eventual time point for the jump in discrete variates, $\text{TCN}_{\text{gate}_3}$ and $\text{TCN}_{\text{gate}_4}$ are gate convolutions used for the continuous and discrete variates, to model the conditional information in the discrete co-ODE branch. Next, $\mathbf{N}(s)$ is the counting function of the Poisson process with the conditional intensity $\lambda(t)$. Finally, $\int_0^t \frac{d\mathbf{Z}(s)}{ds} d\mathbf{N}(s)$ is the compound Poisson process where the discrete hidden state features $\mathbf{Z}(t)$ can jump.

Then, we can obtain the final feature $\mathbf{Z}(T) = \text{ODESolver}\left(\mathbf{Z}(0), \frac{d\mathbf{Z}(t)}{dt}\Big|_{\mathbf{H}(0)}, T; \mathbf{\Theta}_d\right)$ (Jia & Benson, 2019) that learns conditional discrete dynamics.

## 4.5 MULTI-VARIATE PROBABILISTIC MODELING

We use decoders to output the joint probabilities of all observations at time $T$. As Gaussian distributions are the most common continuous probability distributions, our decoder uses fully connected feed-forward layers that outputs an estimated multi-variate Gaussian mixture distribution $N(\hat{\mu}_T, \hat{\Sigma}_T)$ for the observations of the continuous variates at time $T$:

$$\hat{\mu}_T, \hat{\sigma}_T = \text{Decoder}_c\left(\mathbf{H}(T)\right), \tag{8}$$

where $\hat{\mu}_T = (\hat{\mu}_T^1, \hat{\mu}_T^2, \cdots, \hat{\mu}_T^{N_C}) \in \mathbb{R}^{N_C}$ denotes the estimated mean values, $\hat{\sigma}_T = (\hat{\sigma}_T^1, \hat{\sigma}_T^2, \cdots, \hat{\sigma}_T^{N_C}) \in \mathbb{R}^{N_C}$ denotes the estimated standard deviation values that model the stability of different variates, which are used as the diagonal elements of the variance matrix $\hat{\Sigma}_T \in \mathbb{R}^{N_C \times N_C}$ for the $N_C$ continuous variates, respectively. Thus, the MLE in the probabilistic space for the continuous variates is as follows:

$$\mathcal{L}_{MLE} = ln\left[\frac{1}{\sqrt{2\pi|\hat{\Sigma}_T'|}} exp\left(-\frac{1}{2}(\mathbf{c}_T - \hat{\mu}_T)^\top \hat{\Sigma}_T'^{-1}(\mathbf{c}_T - \hat{\mu}_T)\right)\right], \tag{9}$$

where $\hat{\Sigma}_T' = \hat{\Sigma}_T + \epsilon\mathbf{I}$, $\mathbf{I}$ is the unit identity matrix and $\epsilon$ is a small number that makes the diagonal matrix $\hat{\Sigma}_T + \epsilon\mathbf{I}$ a positive-definite matrix to avoid dividing by zero.

To obtain more accurate joint and marginal distributions, we calculate the empirical covariance between the continuous variates and estimate an empirical covariance matrix $\Sigma_T$ to capture their correlations, as different variates are usually dependent of each other:

$$\Sigma_T(j_1, j_2) = \Sigma_T(j_2, j_1) = \frac{1}{T-1}\sum_{i=1}^T (\mathbf{c}_i^{j_1} - \overline{\mathbf{c}}^{j_1})(\mathbf{c}_i^{j_2} - \overline{\mathbf{c}}^{j_2}), \text{ for } 1 \leq j_1, j_2 \leq N_C, \tag{10}$$

where $\overline{\mathbf{c}}^{j_1}$ and $\overline{\mathbf{c}}^{j_2}$ are the mean values of the $j_1$-th and $j_2$-th continuous variates, respectively. Overall, the loss function for the continuous variates is as follows:

$$\mathcal{L}_c = -\mathcal{L}_{MLE} - ln\Big[\frac{1}{\sqrt{2\pi|\Sigma_T'|}}exp\Big(-\frac{1}{2}(\mathbf{c}_T - \hat{\mu}_T)^\top \Sigma_T'^{-1}(\mathbf{c}_T - \hat{\mu}_T)\Big)\Big], \tag{11}$$

where $\Sigma_T' = \Sigma_T + \epsilon\mathbf{I}$. To avoid dividing by zero, a small value $\epsilon$ is used to ensure that matrix $\Sigma_T + \epsilon\mathbf{I}$ is positive-definite. The proof is provided in Appendix B.

As the softmax distribution is the most common discrete probability distribution, our decoder uses fully connected feed-forward layers and outputs the estimated softmax distributions $\hat{\mathbf{d}}_T$ for the observations of discrete variates:

$$\hat{logit}_T^1, \hat{logit}_T^2, \cdots, \hat{logit}_T^{N_D} = \mathrm{Decoder}_d\big(\mathbf{Z}(T)\big), \ \hat{\mathbf{d}}_T^j = \mathrm{softmax}(\hat{logit}_T^j), \ \mathrm{for} j \in \{1, \cdots, N_D\}, \tag{12}$$

where $\hat{\mathbf{d}}_T^j$ denotes the estimated probability distribution for the $j$-th discrete variate like one-hot encoding. The Cross Entropy (CE) loss function for the discrete variates is:

$$\mathcal{L}_d = \mathrm{CE}(\hat{\mathbf{d}}_T, \mathbf{d}_T). \tag{13}$$

Our final loss function is $\mathcal{L} = \mathcal{L}_c + \mathcal{L}_d$, where $\mathcal{L}$ optimizes the model on the embedding layers, $\mathcal{L}_c$ optimizes the model on the continuous co-ODE and $\mathrm{Decoder}_c$, and $\mathcal{L}_d$ optimizes the model on the discrete co-ODE and $\mathrm{Decoder}_d$. During inference, joint probabilities of all data observations denote anomaly scores, *i.e.,* a low joint probability at a timestamp $t_i$ indicates a likely anomaly. As our object function is the Log probability, we compute the object function with data observations and the predicted distributions as the anomaly scores.

## 5 EXPERIMENTS

### 5.1 EXPERIMENTAL SETTINGS

**Datasets.** Following existing time series anomaly detection studies (Jhin et al., 2023; Campos et al., 2022), we use nine real-world datasets from various domains: SMD (Su et al., 2019), PSM (Abdulaal et al., 2021), MSL (Hundman et al., 2018), SMAP (Hundman et al., 2018), SWaT (Xu et al., 2022), GECCO (Lai et al., 2021), SWAN (Lai et al., 2021), CICIDS (Wu & Keogh, 2022) and Credit (Zhao et al., 2022). More details can be seen in Appendix D. We can see that most of these real-world datasets contain both continuous and discrete variates.

**Baselines.** We compare TAD-UP with 21 baselines belonging to different categories. Clustering-based methods: PCA (Liu et al., 2024) and OCSVM (Ruff et al., 2018). Density estimation-based methods: HBOS (Goldstein & Dengel, 2012) DAGMM (Zong et al., 2018), IForest (Hariri et al., 2019), LODA (Pevný, 2016), and LOF (Breunig et al., 2000). Contrastive-based methods: DCdetector (Yang et al., 2023), AnomalyTransformer (A.T.) (Xu et al., 2022), and PAD (Jhin et al., 2023). Forecasting and reconstruction-based methods: AE (Fang et al., 2024), LSTM (Schmidl et al., 2024), OmniAnomaly (Omni) (Su et al., 2019), BeatGAN (Du et al., 2023), CAE-Ensemble (Campos et al., 2022), D3R (Wang et al., 2023), GPT4TS (Zhou et al., 2023), ModernTCN (donghao & wang xue, 2024), MEMTO (Song et al., 2024), and SensitiveHUE (Feng et al., 2024). Foundation model: DADA (Shentu et al., 2025). We note that DADA is a large time series anomaly detection model pre-trained with additional data apart from the datasets listed in section 5.1.

**Metrics.** Recent studies (Wang et al., 2023; Sun et al., 2024; Zhao et al., 2022; Shentu et al., 2025) demonstrate that point adjustments (PA) can lead to faulty performance evaluations. Thus, we use the Affiliation-based Precision (Aff-P), Recall (Aff-R), and F1-score (Aff-F1) (Huet et al., 2022). We also use the ROC-AUC (Paparrizos et al., 2022) metric following DADA, which enables evaluation without choosing a threshold. More details are shown in the Appendix D.2.

### 5.2 MAIN RESULTS AND ANALYSIS

Tables 1 and 2 show the overall anomaly detection performance. We repeat with three different randomly selected seeds for the model initialization and report the average result. The best F1-score and AUC-ROC are highlighted in bold, and the second-best values are underlined. As DADA is a

Table 1: Experiments on five real-world datasets, findings in percentages.

| Dataset | SMD | | | MSL | | | SMAP | | | SWaT | | | PSM | | |
|---|---|---|---|---|---|---|---|---|---|---|---|---|---|---|---|
| Metric | Aff-P | Aff-R | Aff-F1 | Aff-P | Aff-R | Aff-F1 | Aff-P | Aff-R | Aff-F1 | Aff-P | Aff-R | Aff-F1 | Aff-P | Aff-R | Aff-F1 |
| OCSVM | 66.98 | 82.03 | 73.75 | 50.26 | 99.86 | 66.87 | 41.05 | 69.37 | 51.58 | 56.80 | 98.72 | 72.11 | 57.51 | 58.11 | 57.81 |
| PCA | 64.92 | 86.06 | 74.01 | 52.69 | 98.33 | 68.61 | 50.62 | 98.48 | 66.87 | 62.32 | 82.96 | 71.18 | 77.44 | 63.68 | 69.89 |
| HBOS | 60.34 | 64.11 | 62.17 | 59.25 | 83.32 | 69.25 | 41.54 | 66.17 | 51.04 | 54.49 | 91.35 | 68.26 | 78.45 | 29.82 | 43.21 |
| LOF | 57.69 | 99.10 | 72.92 | 49.89 | 72.18 | 59.00 | 47.92 | 82.86 | 60.72 | 53.20 | 96.73 | 68.65 | 53.90 | 99.91 | 70.02 |
| IForest | 71.94 | 94.27 | 81.61 | 53.87 | 94.58 | 68.65 | 41.12 | 68.91 | 51.51 | 53.03 | 99.95 | 69.30 | 69.66 | 88.79 | 78.07 |
| LODA | 66.09 | 84.37 | 74.12 | 57.79 | 95.65 | 72.05 | 51.51 | 100.00 | 68.00 | 56.30 | 70.34 | 62.54 | 62.22 | 87.38 | 72.69 |
| DAGMM | 63.57 | 70.83 | 67.00 | 54.07 | 92.11 | 68.14 | 50.75 | 96.38 | 66.49 | 59.42 | 92.36 | 72.32 | 68.22 | 70.50 | 69.34 |
| A.T. | 54.08 | 97.07 | 66.42 | 51.04 | 95.36 | 66.49 | 56.91 | 96.69 | 71.65 | 53.63 | 98.27 | 69.39 | 54.26 | 82.18 | 65.37 |
| DCdetector | 50.93 | 95.57 | 66.45 | 55.94 | 95.53 | 70.56 | 53.12 | 98.37 | 68.99 | 53.25 | 98.12 | 69.03 | 54.72 | 86.36 | 66.99 |
| PAD | 59.54 | 67.66 | 63.71 | 56.33 | 82.21 | 68.15 | 41.67 | 64.52 | 53.94 | 54.73 | 92.35 | 68.06 | 68.45 | 57.72 | 59.21 |
| AE | 69.22 | 98.48 | 81.30 | 56.66 | 96.66 | 70.72 | 39.42 | 70.31 | 50.52 | 54.92 | 98.20 | 70.45 | 60.67 | 98.24 | 75.01 |
| LSTM | 60.12 | 84.77 | 70.35 | 58.82 | 14.68 | 23.49 | 55.25 | 27.70 | 36.90 | 49.99 | 82.11 | 62.15 | 57.06 | 95.92 | 71.55 |
| BeatGAN | 74.11 | 81.64 | 77.69 | 55.74 | 98.94 | 71.30 | 54.04 | 98.30 | 69.74 | 61.89 | 83.46 | 71.08 | 58.81 | 99.08 | 73.81 |
| Omni | 79.09 | 75.77 | 77.40 | 51.23 | 99.40 | 67.61 | 52.74 | 98.51 | 68.70 | 62.76 | 82.82 | 71.41 | 69.20 | 80.79 | 74.55 |
| CAE-Ensemble | 73.05 | 83.61 | 77.97 | 54.99 | 93.93 | 69.37 | 62.32 | 64.72 | 63.50 | 62.10 | 82.90 | 71.01 | 73.17 | 73.66 | 73.42 |
| D3R | 64.87 | 97.93 | 78.02 | 66.85 | 90.83 | 77.02 | 61.76 | 92.55 | 74.09 | 60.14 | 97.57 | _74.39_ | 73.32 | 88.71 | 80.29 |
| GPT4TS | 73.33 | 95.97 | 83.13 | 64.86 | 95.43 | _77.23_ | 63.52 | 90.56 | _74.67_ | 56.84 | 91.46 | 70.11 | 73.61 | 91.13 | _81.44_ |
| ModernTCN | 74.07 | 94.79 | _83.16_ | 65.94 | 93.00 | 77.17 | 69.50 | 65.45 | 67.41 | 59.14 | 89.22 | 71.13 | 73.47 | 86.83 | 79.59 |
| MEMTO | 49.69 | 98.05 | 65.96 | 52.73 | 97.34 | 68.40 | 50.12 | 96.69 | 66.57 | 56.47 | 98.02 | 71.66 | 52.69 | 83.94 | 64.74 |
| SensitiveHUE | 60.34 | 90.13 | 72.29 | 55.92 | 98.95 | 71.46 | 53.63 | 98.37 | 69.42 | 58.91 | 91.71 | 71.74 | 56.15 | 98.75 | 71.59 |
| DADA | 76.50 | 94.54 | _84.57_ | 68.70 | 91.51 | _78.48_ | 65.85 | 88.25 | _75.42_ | 61.59 | 94.59 | _74.60_ | 74.31 | 92.11 | _82.26_ |
| TAD-UP | 77.26 | 94.53 | **85.03** | 68.81 | 92.37 | **78.87** | 66.49 | 89.13 | **76.16** | 62.51 | 94.87 | **75.36** | 74.32 | 93.18 | **82.68** |

Table 2: Experiments on four real-world datasets with more metrics, findings in percentages.

| Dataset | CICIDS | | Creditcard | | GECCO | | SWAN | |
|---|---|---|---|---|---|---|---|---|
| Metric | Aff-F1 | AUC-ROC | Aff-F1 | AUC-ROC | Aff-F1 | AUC-ROC | Aff-F1 | AUC-ROC |
| A.T. | 34.71 | 49.00 | 65.14 | 52.55 | 64.27 | 51.60 | 33.67 | 44.74 |
| DCdetector | 40.02 | 53.95 | 58.28 | 42.36 | 66.18 | 45.38 | 14.42 | 43.48 |
| ModernTCN | 51.74 | 65.33 | 73.80 | 95.55 | _90.18_ | _95.20_ | 46.45 | _52.63_ |
| GPT4TS | _54.00_ | _67.91_ | _72.88_ | 95.58 | 88.11 | 90.21 | _47.27_ | 51.93 |
| DADA | _73.49_ | _69.33_ | **75.12** | _95.73_ | _90.20_ | _93.44_ | _71.93_ | 53.29 |
| TAD-UP | **75.83** | **72.11** | _74.99_ | **95.80** | **90.32** | 95.28 | 71.98 | **57.42** |

large model pre-trained with additional multi-domain time series data, we also double-underline the second-best ones other than DADA.

Key observations are as follows. First, TAD-UP can achieve state-of-the-art results on almost all datasets. Real-world time series data usually contain different continuous and discrete variates that have different measurement units. This shows that TAD-UP is able to discriminate and learn both continuous and discrete dynamics, which improves the accuracy of anomaly detection on real-world time series data. Second, TAD-UP achieves better accuracy compared to other deep learning-based methods. These methods simply add up all errors from different variates, regardless of their measurement units. Third, TAD-UP outperforms DADA, except for on the Creditcard dataset. This may be because DADA is trained on more financial datasets. In addition, the Creditcard dataset contains only continuous variates, and learning continuous dynamics may yield smaller improvements on financial data. PSM and GECCO datasets also contain only continuous variates. However, as the continuity property are more important in domains of application server and water quality, our method outperforms DADA on PSM and GECCO datasets. Fourth, TAD-UP outperforms PAD on all datasets. PAD can learn only continuous dynamics with NCDEs, which cannot model correlations among different variates neither.

## 5.3 ABLATION STUDY

We conduct an ablation study to assess the components in TAD-UP on four datasets that contains both continuous and discrete variates, *i.e.,* SMD, MSL, SWaT, and CICIDS. Specifically, we compare TAD-UP with the following variants:

- w/o continuous co-ODE: This variant learns features directly from the continuous variates without the co-ODE. It encodes the continuous observations directly with gate TCN to learns hidden features in a discrete timestamp manner like existing TCN-based models, without the ODESolver to learn continuous dynamics. The discrete co-ODE branch and the probabilistic modeling remains.

- w/o discrete co-ODE: This variant learns features directly from the discrete variates without co-ODE. The continuous co-ODE branch and the probabilistic modeling remains.

- w/o co-ODE: This variant learns features from all variates directly using gate TCNs and does not discriminate between continuous and discrete variates.

- w/o probabilistic modeling: This variant uses traditional decoders directly (fully connected feed-forward layers as the decoder to reconstruct time series observations) and leverages the MSE reconstruction errors for training and inference, instead of using our decoders and probabilistic modeling.

- w/o covariance: This variant uses the diagonal variance matrix for probabilistic modeling in Eq. 9, without covariance matrix in Eq. 11. Thus, joint probabilities will treat different variates equally and cannot specifically learn the marginal probability.

- w/o co-ODE and probabilistic modeling: This variant learns features from all variates directly using gate TCNs and does not discriminate between continuous and discrete variates, and it uses the MSE reconstruction errors, instead of using our decoders and probabilistic modeling.

Table 3 shows the results for the different variants: (1) Continuous and discrete variates exhibit different forms of dynamics. Learning and constraining different dynamics accordingly will improve the models and yield more accurate results. (2) Our joint probabilistic modeling helps obtain anomaly scores that consider the importance of different variates with Maximum Likelihood Estimation in the unified probabilistic space, and thus our method is able to obtain more accurate results. (3) The estimated empirical covariances between the continuous variates can help obtain more accurate joint and marginal probability distributions. This also indicates that it is very important to consider the interactions and correlations among multiple variates in time series, as different variates are often dependent. (4) Removing all our major components reduces performance markedly, which fails to discriminate and learn interactive dynamics with different continuous and discrete forms, and cannot model the importance of different variates to obtain the final anomaly scores.

Table 3: Ablation study, findings in percentages

| Dataset | SMD | | | MSL | | | SWaT | | | CICIDS | | |
|---|---|---|---|---|---|---|---|---|---|---|---|---|
| Metric | Aff-P | Aff-R | Aff-F1 | Aff-P | Aff-R | Aff-F1 | Aff-P | Aff-R | Aff-F1 | Aff-P | Aff-R | Aff-F1 |
| w/o continuous co-ODE | 74.96 | 93.20 | 83.09 | 67.98 | 91.34 | 77.95 | 61.94 | 94.67 | 74.88 | 64.55 | 90.26 | 75.27 |
| w/o discrete co-ODE | 75.63 | 92.47 | 83.21 | 67.10 | 91.49 | 77.41 | 61.44 | 93.75 | 74.23 | 61.86 | 90.13 | 73.37 |
| w/o co-ODE | 74.93 | 92.41 | 82.75 | 67.11 | 90.97 | 77.25 | 61.47 | 93.09 | 74.04 | 61.87 | 89.93 | 73.31 |
| w/o probabilistic modeling | 74.85 | 92.93 | 82.92 | 60.01 | 94.66 | 73.45 | 62.37 | 94.91 | 75.27 | 64.37 | 90.52 | 75.24 |
| w/o covariance | 75.72 | 94.36 | 84.02 | 68.62 | 92.39 | 78.76 | 62.40 | 94.72 | 75.23 | 64.61 | 90.89 | 75.52 |
| w/o co-ODE and probabilistic modeling | 74.72 | 92.38 | 82.61 | 58.90 | 93.21 | 72.19 | 60.03 | 89.76 | 71.94 | 60.98 | 88.87 | 72.32 |
| TAD-UP | 77.26 | 94.53 | **85.03** | 68.81 | 92.37 | **78.87** | 62.51 | 94.87 | **75.36** | 64.90 | 91.20 | **75.83** |

## 5.4 MORE EXPERIMENTS

More analytical experiments are presented in Appendix E, including efficiency comparison, parameter sensitivity studies, and irregular time series anomaly detection.

## 6 CONCLUSION

We present TAD-UP, to learn continuous and discrete dynamics for Time series Anomaly Detection via Unified Probabilistic modeling. First, we propose two co-dependent branches of neural ordinary differential equations with the compound Poisson process to learn both continuous and discrete dynamics for different time series variates together. We also propose gate TCNs and empirical covariance matrix in Maximum Likelihood Estimation to model correlations among variates. Second, we propose unified joint probability modeling with multivariate Gaussian distribution and softmax distribution, and optimize our model with the MLE instead of using per-variate reconstruction or contrastive losses. We detect anomalies using joint probabilities that take the importance of different variates into account via marginal distributions. Extensive experiments on nine real-world datasets offer evidence that TAD-UP is capable of state-of-the-art performance. In future research, it is of interest to explore how to improve TAD-UP and NODEs by means of pre-training on the large-scale multi-domain time series data.

## REPRODUCIBILITY STATEMENT

In this work, we have made every effort to ensure reproducibility. In Section 4, from input to output, we provide a detailed description of our method, including model structure, model training, and anomaly scores. In Appendix D.1, we describe all the datasets in detail. In Appendix D.2, we summarize all the default hyper-parameters, including batch size, window size, learning rate, etc. Our codes will be available upon acceptance.

## LIMITATIONS

In this work, we have provided theoretical analysis and experimental results of how continuous and discrete dynamics will mutually reinforce each other in multivariate time series anomaly detection. In future research, it is of interest to explore efficient hyperparameter tuning and pre-training on the large-scale multi-domain time series data. In addition, there may be variates that show both continuous and discrete properties, such as the number of service connections. These variates change in $\mathbb{N}$ as discrete dynamics, and they are also amenable to mathematical calculation as continuous dynamics where slope represents the rate of increase. Even existing open datasets do not include such variates, it is of interesting to model them.

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

Table 4: Notation

| Symbol | Explanation |
|---|---|
| $\mathbf{C} \in \mathbb{R}^{N_C \times T}$ | Time series observations of $N_C$ continuous variates |
| $\mathbf{D} \in \mathbb{N}^{N_D \times T}$ | Time series observations of $N_D$ discrete variates |
| $T$ | Total number of timestamps |
| $\mathbf{c}_i \in \mathbb{R}^{N_C}$ | Observations of continuous variates at timestamp $t_i$ |
| $\mathbf{d}_i \in \mathbb{N}^{N_D}$ | Observations of discrete variates at timestamp $t_i$ |
| $\mathbf{H}(t)$ | Hidden state feature for continuous variates at any continuous time $t$ |
| $\mathbf{Z}(t)$ | Hidden state feature for discrete variates at any continuous time $t$ |
| $\mathcal{H} = \{\tau_{t_i}\}$ | Discrete eventual time points that indicate whether there is a jump at time $t_i$ |
| $\mathbf{N}(t)$ | Counting function for a Poisson process |
| $\lambda(t)$ | Conditional intensity function for a Poisson process |
| $\int_0^t \frac{d\mathbf{Z}(s)}{ds} d\mathbf{N}(s)$ | Compound Poisson process, where the discrete hidden state features $\mathbf{Z}(t)$ can jump |
| $N(\mu, \Sigma)$ | Multi-variate Gaussian distribution with mean vector $\mu$ and covariance matrix $\Sigma$ |

## A  NEURAL DIFFERENTIAL EQUATIONS

Chen et al. (2018) were the first to propose a new paradigm of learning continuous dynamics for continuous variates with ordinary differential equation-based neural networks, which since has been widely studied in time series forecasting (Jin et al., 2023; Jia & Benson, 2019). Unlike traditional time series neural network models that only learn hidden states among discrete timestamps, *e.g.,* recurrent neural networks (RNNs) and TCNs, differential equations can learn the dynamics of the hidden state features $\mathbf{H}(t)$ by integrating along the continuous time $t$ using the differential coefficient $\frac{d\mathbf{H}(t)}{dt}$. Neural ordinary differential equations (NODEs) use the integral directly to learn the hidden state feature $\mathbf{H}(t)$ at any time $t$ as follows:

$$\mathbf{H}(t) = \mathbf{H}(0) + \int_0^t \frac{d\mathbf{H}(s)}{ds} ds, \ \frac{d\mathbf{H}(t)}{dt} = f\big(\mathbf{H}(t); \mathbf{\Theta}\big), \tag{14}$$

where $\mathbf{H}(0)$ is the hidden state feature that denotes the initial condition, and an embedding layer is used to map the input time series into the hidden state feature $\mathbf{H}(0)$. Next, $f(\cdot; \mathbf{\Theta})$ denotes a neural network with learnable parameters $\mathbf{\Theta}$ that estimates the differential coefficient for continuous hidden state feature $\mathbf{H}(t)$, *i.e.,* the derivative $\frac{d\mathbf{H}(t)}{dt}$. Then, with the ODESolver$(\cdot)$ algorithm (Jia & Benson, 2019) we can obtain the continuous features $\mathbf{H}(t)$ at any time $t$ that can be used in downstream tasks:

$$\mathbf{H}(t) = \text{ODESolver}\big(\mathbf{H}(0), f, t; \mathbf{\Theta}\big). \tag{15}$$

To improve the performance and robustness of the original NODEs, neural controlled differential equations (NCDEs) have been proposed recently. They utilize the Riemann-Stieltjes integral (Kidger et al., 2020; Jhin et al., 2023) to control the continuous dynamics of hidden state feature $\mathbf{H}(t)$ at any time $t$ with time series observations $\mathbf{C}$ as follows:

$$\mathbf{H}(t) = \mathbf{H}(0) + \int_0^t \frac{d\mathbf{H}(s)}{ds} d\mathbf{C}(s) = \mathbf{H}(0) + \int_0^t f\big(\mathbf{H}(s); \mathbf{\Theta}\big) \frac{d\mathbf{C}(s)}{ds} ds, \tag{16}$$

where $\mathbf{C}(t)$ is a continuous time series path across time, as obtained by interpolation from the original time series observations $\mathbf{C}$ with natural cubic spline interpolation (Kidger et al., 2020; Jhin et al., 2023). Finally, $\frac{d\mathbf{C}(s)}{ds}$ is the differential coefficient of the time series path.

Traditional NODEs only use the embedding layers to map the input time series observations $\mathbf{C}$ into the initial hidden state feature $\mathbf{H}(0)$, and the following continuous dynamics $\mathbf{H}(t)$ is learned by integral on $\frac{d\mathbf{H}(t)}{dt}$ with neural network $f(\cdot)$ and parameters $\mathbf{\Theta}$ implicitly. The difference between

NODEs and NCDEs is the continuous time series path $C(t)$, whose differential coefficient controls the continuous dynamics explicitly. However, NCDEs have an inefficient recurrent architecture, and the natural cubic spline interpolation incurs additional space and time complexity.

Neither existing NODEs nor NCDEs can learn the discrete dynamics of discrete variates, where observations can jump between consecutive timestamps. For example, a computing system can run in two different modes 0 and 1. When switching the running modes, the discrete variate jumps immediately between 0 and 1. But existing methods will still output meaningless values for running mode such as 0.1 that do not make sense. Further, they cannot model the correlations between observations from continuous versus discrete variates.

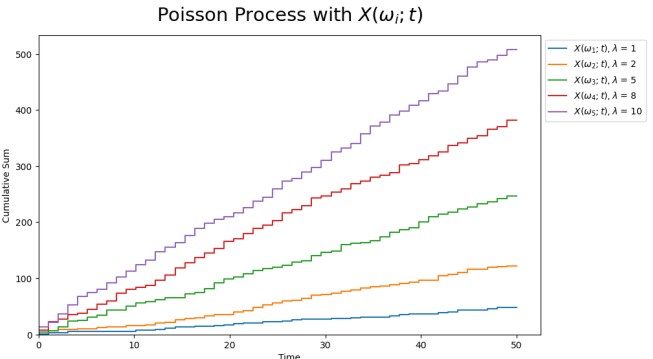

Figure 3: The counting functions of Poisson processes with different intensities $\lambda$.

## B  THEOREMS AND PROOFS

**Theorem 1:**  The covariance matrix is positive semi-definite.

**Proof:**  Let $\mathbf{X} \in \mathbb{R}^{n \times d}$ be a data matrix, where each of the $n$ rows is an observation and each column is a random variate. Without loss of generality, assume that the data is centered, *i.e.,* the mean of each column is zero. The sample covariance matrix is defined as:

$$\Sigma = \frac{1}{n}\mathbf{X}^\top \mathbf{X} \tag{17}$$

We want to show that $\Sigma$ is positive semi-definite, *i.e.,* for any vector $\mathbf{z} \in \mathbb{R}^d$:

$$\mathbf{z}^\top \Sigma \mathbf{z} \geq 0 \tag{18}$$

Proof:

$$\mathbf{z}^\top \Sigma \mathbf{z} = \mathbf{z}^\top \left( \frac{1}{n}\mathbf{X}^\top \mathbf{X} \right) \mathbf{z} = \frac{1}{n}\mathbf{z}^\top \mathbf{X}^\top \mathbf{X}\mathbf{z}$$
$$= \frac{1}{n}(\mathbf{X}\mathbf{z})^\top (\mathbf{X}\mathbf{z}) = \frac{1}{n}\|\mathbf{X}\mathbf{z}\|^2 \geq 0 \tag{19}$$

Since the squared L-2 norm $\|\mathbf{X}\mathbf{z}\|^2$ is always non-negative, it follows that:

$$\mathbf{z}^\top \Sigma \mathbf{z} \geq 0 \quad \forall \mathbf{z} \in \mathbb{R}^d \tag{20}$$

Therefore, the covariance matrix $\Sigma$ is positive semi-definite. $\square$

**Theorem 2:**  The sum of a positive semi-definite matrix and a positive definite matrix is positive definite.

**Proof:**  Let $A \in \mathbb{R}^{n \times n}$ be a positive semi-definite matrix, and let $B \in \mathbb{R}^{n \times n}$ be a positive definite matrix. We claim that the matrix $A + B$ is positive definite. By definition:

- For all non-zero vectors $\mathbf{x} \in \mathbb{R}^n$, we have

$$\mathbf{x}^\top A \mathbf{x} \geq 0$$

- For all non-zero vectors $\mathbf{x} \in \mathbb{R}^n$, we have

$$\mathbf{x}^\top B \mathbf{x} > 0$$

Now consider the matrix $C = A + B$. For any non-zero vector $\mathbf{x} \in \mathbb{R}^n$, we have:

$$\mathbf{x}^\top C \mathbf{x} = \mathbf{x}^\top (A + B)\mathbf{x} = \mathbf{x}^\top A \mathbf{x} + \mathbf{x}^\top B \mathbf{x} \qquad (21)$$

Since $\mathbf{x}^\top A \mathbf{x} \geq 0$ and $\mathbf{x}^\top B \mathbf{x} > 0$, we get:

$$\mathbf{x}^\top C \mathbf{x} > 0 \qquad (22)$$

Therefore, $C = A + B$ is positive definite. $\square$

## C  THEORETICAL ANALYSIS

In this section, we provide theoretical local sensitivity analysis of how continuous and discrete dynamics mutually reinforce each other in multivariate time series anomaly detection.

### C.1  LOCAL SENSITIVITY OF OUR CO-DEPENDENT CONDITIONAL ODEs

**Setup:**

The continuous hidden state $H(t) \in \mathbb{R}^n$ is solved from the ODE

$$\left. \frac{dH(t)}{dt} \right|_{Z(0)} = f\big(H(t), Z(0); \boldsymbol{\Theta}_c\big), \qquad H(0) \text{ initialized by continuous variates.}$$

Our $f$ is based on the TCN layer which is a $C^1$ differentiable function, and for each different $Z(0)$ initialized by discrete variates the ODE problem is well posed (unique solution). Denote partial derivatives (Jacobians) by

$$A(t) := \frac{\partial f}{\partial H}\big(H(t), Z(0)\big) \in \mathbb{R}^{n \times n}, \qquad B(t) := \frac{\partial f}{\partial Z}\big(H(t), Z(0)\big) \in \mathbb{R}^{n \times n}.$$

where $A(t), B(t)$ are continuous on $[0, T]$ and bounded.

The discrete hidden state $Z(t) \in \mathbb{R}^n$ is solved from the intensity $\lambda(t) = \mathrm{FC}_\lambda\big(\mathbf{Z}(t)\big)$ and the compound Poisson process ODE with

$$\left. \frac{dZ(t)}{dt} \right|_{H(0)} = g\big(Z(t), H(0); \boldsymbol{\Theta}_d\big), \qquad Z(0) \text{ initialized by discrete variates.}$$

Each jump will update the hidden state $Z(t)$ with $g(Z(t), H(0))$. For the sensitivity analysis below we primarily consider the mean dynamics.

**Local sensitivity of $H(T)$:**

Consider small perturbations $\delta H(0) \in \mathbb{R}^n$ and $\delta Z(0) \in \mathbb{R}^n$ in the initial condition. Let $\delta H(t)$ denote the first-order variation (Gateaux derivative) of the solution with respect to these perturbations. Linearizing our continuous ODE gives the variational equation

$$\frac{d}{dt} \delta H(t) = A(t)\, \delta H(t) + B(t)\, \delta Z(0).$$

This is a non-homogeneous linear ODE. Let $\Phi(t, s) \in \mathbb{R}^{n \times n}$ be the state transition matrix solving

$$\frac{\partial}{\partial t} \Phi(t, s) = A(t)\, \Phi(t, s), \qquad \Phi(s, s) = I.$$

Then the solution of the variational equation is

$$\delta H(T) = \Phi(T, 0)\, \delta H(0) + \int_0^T \Phi(T, s)\, B(s)\, ds\ \delta Z(0). \qquad (\text{S1})$$

The first term $\Phi(T, 0)\delta H(0)$ is the usual sensitivity of continuous dynamics to initial continuous perturbations, as in existing NODEs (Chen et al., 2018; Westny et al., 2024). The second term is an explicit linear mapping from the *initial discrete perturbation* $\delta Z(0)$ to the final continuous variation $\delta H(T)$. As the matrix integral $\int_0^T \Phi(T, s) B(s)\, ds$ is non-zero (which holds as $B(s) \not\equiv 0$), the change in the initial discrete condition $Z(0)$ will change the continuous hidden state $H(T)$ accordingly.

**Local sensitivity of $Z(T)$:**

The analysis for discrete branch is under the mean dynamics. The counting process $N(t)$ has intensity $\lambda(t) = \mathrm{FC}_\lambda\big(\mathbf{Z}(t)\big)$, and thus we have that next possible jump will cause an expected update $\mathbb{E}[\Delta Z] = g(Z(t), H(0))\,\lambda(t)\,\Delta t$. Then, in mean-field form, the expected discrete hidden state $m(t) := \mathbb{E}[Z(t)]$ satisfies approximately

$$\frac{d}{dt}m(t) = \lambda(t)\, g(Z(t), H(0)),$$

where $g(Z(t), H(0))$ is a bounded function depending on $H(0)$.

Linearizing the mean equation gives the first-order variation $\delta m(t)$:

$$\frac{d}{dt}\delta m(t) \approx \lambda(t)\left(\frac{\partial g}{\partial Z}\right)(Z(t), H(0))\,\delta Z(t) + \lambda(t)\left(\frac{\partial g}{\partial H}\right)(Z(t), H(0))\delta H(0).$$

Let $\Psi(t, s)$ be its state transition matrix as defined above. As $\lambda(t)$ is bounded, we have

$$\delta\mathbb{E}[Z(t)] \approx \Psi(T, 0)\,\delta Z(0) + \int_0^T \Psi(T, s)\left(\frac{\partial g}{\partial H}\right)(Z(s), H(0)\, ds \ \delta H(0). \tag{S2}$$

The first term $\Psi(T, 0)\delta Z(0)$ is the discrete sensitivity to initial discrete perturbations. The second term is an explicit linear mapping from the *initial continuous perturbation* $\delta Z(0)$ to the final discrete variation $\delta H(T)$. Thus, the change in the initial continuous condition $H(0)$ will change the discrete hidden state $Z(T)$ accordingly.

## C.2 REINFORCE IN TIME SERIES ANOMALY DETECTION:

The improvement of our conditional continuous dynamics $\left.\frac{dH(t)}{dt}\right|_{Z(0)}$ and conditional discrete dynamics $\left.\frac{dZ(t)}{dt}\right|_{H(0)}$ is from the mutual information: a higher mutual information concentrates the joint density of normal samples and therefore makes anomalous samples more separable.

The mutual information of $C$ and $D$, where $C$ denotes continuous variates and $D$ denotes discrete variates, is given by:

$$I(C; D) = D_{\mathrm{KL}}\big(p(c, d)\,\big\|\,p(c)p(d)\big) = E(C) - E(C \mid D) = E(D) - E(D \mid C), \tag{23}$$

where $E(\cdot)$ denotes entropy. As usually holds that continuous and discrete variates are correlated, we have $I(C; D) > 0$, $E(C) > E(C \mid D)$, and $E(D) > E(D \mid C)$.

For continuous variates $C$ with density $p(c)$, the traditional time series anomaly detection methods assume that normal samples lie within a small set $\mathcal{T}_C$ that has large probability density whose volume satisfies

$$\mathrm{Vol}(\mathcal{T}_C) \approx \exp\big(E(C)\big). \tag{24}$$

When our method models the conditional dynamics, the conditional distributions $p(c \mid d)$ and $p(d \mid c)$ will be sharper than $p(c)$ and $p(d)$, as $E(C) > E(C \mid D)$, and $E(D) > E(D \mid C)$. Thus, conditioning on $D$ yields a smaller set $\mathcal{T}_{C|D}$ that has larger probability density:

$$\mathrm{Vol}(\mathcal{T}_{C|D}) \approx \exp\big(H(C \mid D)\big), \tag{25}$$

and the ratio of volumes satisfies

$$\frac{\mathrm{Vol}(\mathcal{T}_{C|D})}{\mathrm{Vol}(\mathcal{T}_C)} = \exp(H(C \mid D) - H(C)) = \exp(-I(C; D))\,. \tag{26}$$

The decision boundary between normal and abnormal data has been improved. Thus, the mutual reinforcement between the continuous and discrete dynamics enhances anomaly detection.

# D EXPERIMENTAL DETAILS

## D.1 DATASETS

Table 5: Benchmark dataset statistics.

| Dataset | Domain | $N_C$ | $N_D$ | #Training (Unlabeled) | #Validation (Unlabeled) | #Test (Labeled) | Anomaly Rate (%) |
|---------|--------|-------|-------|----------------------|------------------------|-----------------|------------------|
| SMD | server machine | 33 | 5 | 566,724 | 141,681 | 708,420 | 4.16 |
| MSL | NASA space sensor | 1 | 54 | 46,653 | 11,663 | 73,729 | 10.72 |
| SMAP | NASA space sensor | 1 | 24 | 108,146 | 27,036 | 427,617 | 13.13 |
| SWaT | water treatment | 25 | 26 | 396,000 | 99,000 | 449,919 | 12.14 |
| SWAN | space solar weather | 33 | 5 | 48,000 | 12,000 | 60,000 | 32.6 |
| CICIDS | web server | 58 | 14 | 68,092 | 17,023 | 85,116 | 1.28 |
| PSM | application server | 25 | 0 | 105,885 | 26,398 | 87,841 | 27.8 |
| GECCO | water quality | 9 | 0 | 55,408 | 13,852 | 69,261 | 1.1 |
| Creditcard | finance | 29 | 0 | 159,491 | 39,873 | 85,443 | 0.172 |

The datasets are representative open data for multi-variate time series anomaly detection evaluation.

- SMD (Su et al., 2019) (Server Machine Dataset) is collected from a computing cluster, where different server machines record resource utilization yielding 38 dimensions.
- PSM (Abdulaal et al., 2021) (Pooled Server Metrics) is a 25-dimensional dataset collected by eBay that shows the performance of multiple application servers.
- MSL (Hundman et al., 2018) (Mars Science Laboratory Rover) is from the spacecraft monitoring systems and is collected by NASA. It includes the health check-up data from Mars rover sensors.
- SMAP (Hundman et al., 2018) (Soil Moisture Active Passive) is a 25-dimension dataset collected by NASA, which contains soil samples and telemetry data.
- SWaT (Xu et al., 2022) (Secure Water Treatment) is a 51-dimensional dataset collected from sensors of the critical infrastructure systems under continuous operations for secure water treatment.
- NeurIPS-TS (Lai et al., 2021) includes CICIDS, Creditcard, GECCO, and SWAN from different domains. CICIDS is a 62-dimensional dataset collected from multiple web servers. GECCO is a 9-dimensional dataset collected from cyber-physical systems monitoring the drinking water quality. SWAN is a 38-dimensional dataset extracted from solar photospheric vector magnetograms for space weather. Creditcard is a 29-dimensional dataset from the finance domain.

Statistical information is given in Table 5, where $N_C$ and $N_D$ are the number of continuous and discrete variates observed at each timestamp, respectively. We can see that most of these real-world datasets contain both continuous and discrete variates. For datasets containing only continuous variables we simply remove the corresponding discrete co-ODE branch, $TCN_{gate_2}$, and cross entropy branch. We follow the same training-validation-test splits as in the previous studies (Shentu et al., 2025; Xu et al., 2022; Zhao et al., 2022), shown in the table.

## D.2 IMPLEMENT DETAILS

Our code will be made publicly available upon acceptance. First, we report the experimental results from the original papers of each baseline if we have the same experimental settings. For the rest experimental results, We employ the official open-source implementations of baseline methods and have carefully tuned their hyperparameters based on the recommendations from their papers. The experiments are conducted on an Ubuntu 18.04.5 LTS system server and Pytorch 1.2.0, with Intel(R) Xeon(R) Gold 5215 CPU @ 2.50GHz, NVIDIA Tesla-A800-80GB GPU and NVIDIA Quadro RTX

8000 GPU. Our method uses the Adam optimizer with the default parameters and the learning rate set to 0.0001 by default. The sliding window size is tuned among $\{16, 32, 64\}$. The batch size is tuned among $\{32, 64\}$. The dimensionality of the hidden features, *i.e.,* $d$, is tuned among $\{16, 32, 64\}$. We use three different kernel sizes for TCNs, *i.e.,* $k = \{2, 3, 5\}$. The dilation factor $r$ is set to 1 by default. Following the existing settings in state-of-the-art methods (Wang et al., 2023; Shentu et al., 2025; Zhao et al., 2022), we use the same strategy, called SPOT, to choose the threshold of anomaly scores for all methods when computing binary anomaly labels and metric values for the metrics Aff-Precision, Aff-Recall, and Aff-F1. The ROC-AUC metric does not require a threshold as it considers the average accuracy under all possible thresholds.

## E    MORE EXPERIMENTS

### E.1    VISUALIZATIONS

We provide further visualizations to explain how continuous and discrete dynamics mutually enhance multivariate time series anomaly detection, with the real-world SWaT dataset.

From Figure 4(a), we can observe that continuous and discrete variates are correlated. From Figure 4(b), we can observe that the discrete variates are anomalies conditional on other variates. From Figures 4(c) and (d), we can observe that the decision boundary between normal and abnormal data has been improved with our method, which is consistent with our theoretical analysis in Appendix C.2.

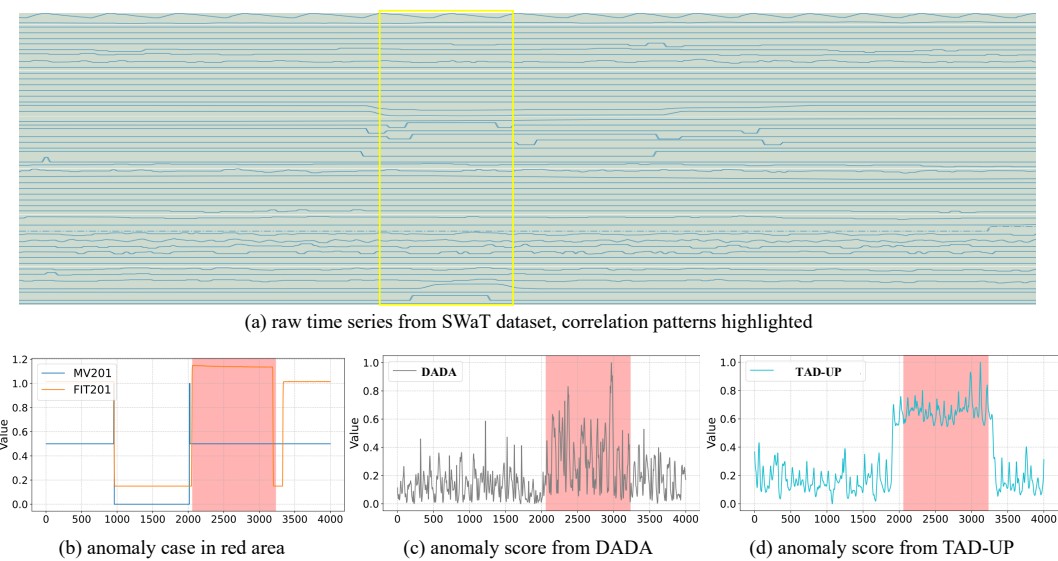

(a) raw time series from SWaT dataset, correlation patterns highlighted

(b) anomaly case in red area     (c) anomaly score from DADA     (d) anomaly score from TAD-UP

Figure 4: Visualizations with the real-world SWaT dataset.

### E.2    EFFICIENCY

We evaluate the efficiency-accuracy tradeoff for our method in terms of inference time and F1-score, compared to the state-of-the-art baselines. The results are reported in Table 6.

Inference time refers to the time required for anomaly detection on the batch of testing samples. We see that TCN-based methods, *i.e.,* ModernTCN and DADA, are more efficient than RNN or Transformer-based methods, *i.e.,* A.T. and GPT4TS. We also see that our method performs the best with respective to the efficiency-accuracy tradeoff, as TCN backbones are more efficient than RNN or Transformer backbones, and as NODEs are are more efficient than NCDEs.

Table 6: Efficiency and accuracy

| Dataset | SMD | | SWaT | | PSM | |
|---|---|---|---|---|---|---|
| Method | inference time (s/iteration) | accuracy (Aff-F1) | inference time (s/iteration) | accuracy (Aff-F1) | inference time (s/iteration) | accuracy (Aff-F1) |
| A.T. | 0.093 | 66.42 | 0.118 | 69.39 | 0.065 | 65.37 |
| PAD | 0.088 | 63.71 | 0.092 | 68.06 | 0.084 | 59.21 |
| ModernTCN | 0.038 | 83.16 | 0.060 | 71.13 | 0.036 | 79.59 |
| GPT4TS | 0.113 | 83.13 | 0.139 | 70.11 | 0.093 | 81.44 |
| DADA | 0.081 | 84.57 | 0.097 | 74.60 | 0.075 | 82.26 |
| TAD-UP | 0.054 | **85.03** | 0.058 | **75.36** | 0.050 | **82.68** |

Table 7: Parameter sensitivity, window size.

| Dataset | SWaT | | | | PSM | | | | CICIDS | | | |
|---|---|---|---|---|---|---|---|---|---|---|---|---|
| window size | Aff-P | Aff-R | Aff-F1 | AUC-ROC | Aff-P | Aff-R | Aff-F1 | AUC-ROC | Aff-P | Aff-R | Aff-F1 | AUC-ROC |
| 8 | 62.03 | 94.15 | 74.79 | 82.33 | 73.45 | 92.41 | 81.84 | 60.67 | 64.41 | 91.00 | 75.43 | 71.85 |
| 16 | 62.19 | 94.36 | 74.97 | 82.72 | 74.32 | 93.18 | **82.62** | **61.35** | 64.90 | 91.20 | **75.83** | **72.01** |
| 32 | 62.51 | 94.87 | **75.36** | **83.01** | 73.44 | 92.97 | 82.06 | 60.70 | 64.22 | 91.06 | 75.32 | 70.03 |
| 64 | 62.36 | 94.47 | 75.13 | 82.93 | 72.58 | 91.76 | 81.05 | 60.19 | 64.15 | 90.53 | 75.43 | 70.57 |

## E.3 PARAMETER SENSITIVITY STUDIES

We evaluate the impact of the sliding window sizes in Tables 7 and 8. We can see that TAD-UP is relatively unaffected by the sliding window sizes. We can also see that time series from different domains may require different sliding window sizes for optimal window lengths. For example, window size 32 performs best on SWaT, MSL, and GECCO, and window size 16 performs best on PSM, CICIDS, and SMAP. This may be because different application domains have different sampling intervals and different time series have different frequencies.

We evaluate the impact of the kernel sizes in the TCN backbones. The results are reported in Table 10 show that combining different kernel sizes and pooling different hidden features improves the results. This can be also seen in previous TCN-based methods, where the use of multiple kernel sizes enables the learning of different temporal features with different periodic information from time series.

## E.4 IRREGULAR TIME SERIES

PAD (Jhin et al., 2023) uses interpolation and NCDEs to support anomaly detection on time series with irregular timestamps. We also show the experimental results on the irregular time series, by dropping observations at random timestamps and learning at the remaining timestamps as in the study of PAD (Jhin et al., 2023). Table 9 shows that TAD-UP still performs well for irregular time series as the co-ODEs are able to learn dynamics along continuous time.

Table 8: Parameter sensitivity, window size.

| Dataset | MSL | | | | SMAP | | | | GECCO | | | |
|---|---|---|---|---|---|---|---|---|---|---|---|---|
| window size | Aff-P | Aff-R | Aff-F1 | AUC-ROC | Aff-P | Aff-R | Aff-F1 | AUC-ROC | Aff-P | Aff-R | Aff-F1 | AUC-ROC |
| 8 | 66.82 | 91.34 | 77.17 | 66.54 | 66.03 | 88.47 | 75.62 | 50.11 | 82.35 | 97.58 | 89.32 | 94.07 |
| 16 | 67.99 | 92.36 | 78.19 | 66.63 | 66.49 | 89.13 | **76.16** | **50.22** | 83.16 | 98.34 | 90.11 | 94.78 |
| 32 | 68.81 | 92.37 | **78.87** | **68.92** | 66.10 | 89.11 | 75.89 | **50.22** | 83.31 | 98.62 | **90.32** | **95.28** |
| 64 | 66.95 | 92.31 | 77.53 | 67.10 | 65.92 | 89.03 | 75.75 | 50.02 | 83.02 | 97.66 | 89.74 | 95.13 |

Table 9: Anomaly detection for irregular time series.

| Dataset | SWaT | | | | | | | | | | | |
|---|---|---|---|---|---|---|---|---|---|---|---|---|
| Dropping ratio | 0% | | | | 20% | | | | 40% | | | |
| Metric | Aff-P | Aff-R | Aff-F1 | AUC-ROC | Aff-P | Aff-R | Aff-F1 | AUC-ROC | Aff-P | Aff-R | Aff-F1 | AUC-ROC |
| PAD | 54.73 | 92.35 | 68.06 | 63.86 | 52.71 | 92.13 | 67.05 | 63.91 | 49.73 | 90.64 | 64.22 | 56.84 |
| TAD-UP | **62.51** | **94.87** | **75.36** | **83.01** | **61.82** | **93.06** | **74.28** | **80.92** | **59.98** | **91.97** | **72.60** | **72.90** |

Table 10: Parameter sensitivity, kernel sizes in TCNs.

| Dataset | SWaT | | | | PSM | | | | CICIDS | | | |
|---|---|---|---|---|---|---|---|---|---|---|---|---|
| kernel size | Aff-P | Aff-R | Aff-F1 | AUC-ROC | Aff-P | Aff-R | Aff-F1 | AUC-ROC | Aff-P | Aff-R | Aff-F1 | AUC-ROC |
| {2} | 61.80 | 94.08 | 74.57 | 82.64 | 74.18 | 93.12 | 82.57 | 61.24 | 64.02 | 91.06 | 75.18 | 71.67 |
| {3} | 62.53 | 94.01 | 75.10 | 82.61 | 73.95 | 92.87 | 82.33 | 61.23 | 64.50 | 90.83 | 75.43 | 71.99 |
| {5} | 62.11 | 94.58 | 74.98 | 82.53 | 73.85 | 92.95 | 82.31 | 61.29 | 64.47 | 91.02 | 75.47 | 71.93 |
| {2,3,5} | 62.51 | 94.87 | **75.36** | **83.01** | 74.32 | 93.18 | **82.62** | **61.35** | 64.90 | 91.20 | **75.83** | **72.01** |

