# OpenReview forum: "Learning Continuous and Discrete Dynamics for Time Series Anomaly Detection via Probabilistic Modeling"
_ICLR.cc/2026/Conference — Submitted to ICLR 2026_

### Official Review · Reviewer_NV9B · 2025-10-26

**Soundness:** 2
**Presentation:** 1
**Contribution:** 2
**Rating:** 2
**Confidence:** 4

**Summary:**

This paper proposes a probabilistic framework for multivariate time-series anomaly detection that separately learns continuous and discrete dynamics through two co-dependent neural ODE branches. Experiments on nine datasets show competitive or state-of-the-art performance.

**Strengths:**

The idea of separately learning continuous and discrete dynamics and integrating them through co-dependent neural ODEs is interesting and offers a new perspective for handling heterogeneous multivariate time series.

**Weaknesses:**

1. Lack of design motivation. The paper introduces several components, continuous co-ODE, discrete co-ODE, and gated TCN, but does not sufficiently explain why these particular designs are chosen.
2. Limited novelty in model architecture. The proposed framework appears to be a direct combination of existing elements: standard embeddings for continuous/discrete variables, TCNs for temporal modeling, and ODE formulations. The overall contribution lies more in integration than in new methodological innovation.
3. Modeling of discrete dynamics. The discrete branch relies on a Poisson-process-based formulation (Eq. (5)), which inherently models binary “jump/no-jump” events. It is unclear how this approach generalizes to multi-valued discrete states, which are common in categorical time-series data.
4. Expressiveness of Gaussian mixture assumption. The joint probability of all variables is modeled by a (mixture) Gaussian distribution, whose representational capacity is limited for complex, high-dimensional time series.
5. Loss formulation ambiguity. In Eq. (11), the latter term of L_c seems to include  the L_{MLE} term already present in the first half, raising doubts about whether both parts are necessary or if redundancy exists.
6. Weak theoretical contribution. Theorems 1 and 2 in the Appendix are standard linear-algebra results (positive semi-definiteness of covariance matrices) and do not provide theoretical insight specific to the proposed model.
7. Unclear inference mechanism. The paper emphasizes that marginal probabilities help identify important variables in anomaly scoring, but the inference section does not explain how this importance is actually utilized or interpreted during detection.
8. Missing baselines. The experimental comparison omits recent diffusion-based methods such as ImDiffusion, which are relevant and often strong baselines for time-series anomaly detection.

**Questions:**

See the weaknesses section.

---

> ### Author Response · Authors · 2025-11-25
> **Response to Reviewer NV9B (Part I)**
>
> We would like to sincerely thank Reviewer **NV9B** for the comments to improve our paper. We have revised our paper with blue color accordingly.
>
> **W1: Motivation**
>
>  **A:** We would like to clarify that we have explained our motivation in Sections 1, and 4.3 and 4.4.
>
> Continuous and discrete dynamics behave very differently over time. Breaking these **continuity or discreteness property** can indicate higher possibility of the abnormal. However, all existing methods remain unable to discriminate and learn such continuous and discrete dynamics, limiting their effectiveness in multivariate time series anomaly detection. Thus, we propose continuous co-ODE and discrete co-ODE to constrain the continuity and discreteness property, respectively, to improve anomaly detection, as stated in Sections 1.
>
> Variates with continuous and discrete dynamics may be correlated. Breaking these **correlation patterns** can indicate higher possibility of the abnormal. However, existing methods fail to model such correlations across continuous and discrete dynamics. Thus, we propose gated TCN to model the correlations among discrete and continuous variates, and use TCN as the backbone as it is more **efficient** than RNN and Transformer, as stated in Sections 4.3 and 4.4.
>
>
> **W2: Novelty**
>
>  **A:** We would like to clarify that we have summarized our contributions in Section 1 and emphasize that the overall design is not a simple combination of existing elements. In addition, we would like to clarify that we propose a novel design encompassing gated co-dependent NODEs with compound Poisson process to learn correlated continuous and discrete dynamics, instead of using the existing TCN and ODE formulations that can only learn continuous dynamics.
>
> **W3: Poisson process**
>
>  **A:** We would like to clarify that we have introduced the discrete branch in line 285. Our discrete branch is based on a **compound** Poisson process, not a simple Poisson process. **Compound** Poisson process is able to model multi-valued discrete states.
> Specifically, the counting function $ \mathbf{N} ( s ) $ controls the binary “jump/no-jump”,
> and our compound Poisson process $ \int _{0} ^{t} \frac{ \displaystyle d \mathbf{Z} ( s )}{\displaystyle ds}d \mathbf{N} ( s )$ controls the changes $\displaystyle d \mathbf{Z} ( s )$ between multi-valued states.
>
> **W4: Gaussian capacity**
>
>  **A:** Gaussian assumption is commonly used in time series modeling, such as Gaussian Process, and the empirical covariance matrix enables high capacity to model correlations for complex high-dimensional time series. We also evaluated our method with nine commonly used datasets, including CICIDS dataset that contains complex 72-dimensional time series. Our method achieves relative accuracy improvements up to 3% on the AUC-ROC metric even compared with the large foundation model DADA.
>
> **W5: Loss formulation**
>
>  **A:** We would like to clarify that we have introduced both parts of the losses in lines 311 and 319. The first half, i.e., $ \mathcal{L} _ {MLE}$, models the stability of different variates but cannot capture the empirical correlations. In addition, the second half capture the empirical correlations. Thus, both parts are necessary for our loss function $\mathcal{L} _ {c}$.
> To evaluate the importance of capturing the empirical correlations, we have conducted the ablation study **w/o covariance** in the original submission. This variant uses the diagonal variance matrix for probabilistic modeling in Eq. 9, without covariance matrix in Eq. 11. From Table 3, we can observe that our loss function $\mathcal{L} _ {c}$ with both parts performs better compared to **w/o covariance**.

---

> ### Author Response · Authors · 2025-11-25
> **Response to Reviewer NV9B (Part II)**
>
> **W6: Theoretical contribution**
>
>  **A:** We have revised our paper in Appendix C, including more theoretical analysis.
>
> **First**, our method can model the conditional continuous and discrete dynamics $\frac{dH(t)}{dt}\Big| _{Z(0)}$ and $\frac{dZ(t)}{dt}\Big| _{H(0)}$. More theoretical analysis is provided in Appendix C.1.
>
> **Second**, the mutual reinforcement of our conditional continuous and discrete dynamics is resulted from the mutual information: a higher mutual information concentrates the joint density of normal samples and therefore makes anomalous samples more separable (this theoretical analysis is provided in Appendix C.2 and below).
>
> The mutual information of $C$ and $D$, where $C$ denotes continuous variates
> and $D$ denotes discrete variates, is given by:
> $I(C;D)=KL\big(p(c,d)\big\|p(c)p(d)\big)= E(C) - E(C\mid D)= E(D) - E(D\mid C),$
> where $E(\cdot)$ denotes entropy. As usually holds that continuous and discrete variates are correlated, we have $I(C;D) > 0$, $E(C) > E(C\mid D)$, and $E(D) > E(D\mid C)$.
>
> For continuous variates $C$ with density $p(c)$, the traditional time series anomaly detection methods assume that normal samples lie
> within a small set $\mathcal{T}_C$ that has large probability density whose volume satisfies
> $ \mathrm{Vol} ( \mathcal{T}_C ) \approx \exp\big(E(C)\big).$
>
> When our method models the conditional dynamics, the conditional distributions
> $p(c\mid d)$ and $p(d\mid c)$ will be sharper than $p(c)$ and $p(d)$, as $E(C) > E(C\mid D)$, and $E(D) > E(D\mid C)$. Thus, conditioning on $D$ yields a smaller set $ \mathcal{T} _ {C \mid D}$ that has larger probability density:
> $ \mathrm{Vol} ( \mathcal{T} _ {C \mid D} ) \approx
> \exp\big( H ( C \mid D ) \big),$
> and the ratio of volumes satisfies
> $\frac{\mathrm{Vol}(\mathcal{T}_{C\mid D})}{\mathrm{Vol}(\mathcal{T}_C)}
> =\exp\!\left(H(C\mid D)-H(C)\right)=\exp\!\left(-I(C;D)\right).$
>
> The decision boundary between normal and abnormal data has been improved.
> Thus, the mutual reinforcement between the continuous and discrete dynamics enhances anomaly detection.
>
> **W7: Inference and margin probability**
>
>  **A:** We have revised our paper in line 344 as follows to make inference stage more clear: To align with existing time series anomaly detection methods, our problem definition and experimental settings are also detecting time series anomaly at each timestamp $t_i$. During inference, joint probabilities of all data observations, which take the marginal probabilities of different variates into account, denote the anomaly scores, i.e., a low joint probability at a timestamp $t_i$ indicates a likely anomaly. As our object function is the Log probability, we compute the object function with data observations and the predicted distributions as the anomaly scores.
>
> To evaluate the importance of marginal probabilities, we have conducted the ablation study **w/o covariance** in the original submission. This variant uses the diagonal variance matrix for probabilistic modeling in Eq. 9, without covariance matrix in Eq. 11. Thus, joint probabilities will treat different variates equally, and the model cannot specifically learn the marginal probability. From Table 3, we can observe that our method performs better compared to **w/o covariance**, as our method can capture more accurate marginal probabilities with covariance matrix.
>
> **W8: Missing baselines**
>
> **A:** We would like to clarify that we have included state-of-the-art diffusion-based method D3R as our baseline in Table 1 in our original submission. Further, as you suggested, we add more experimental results with ImDiffusion. We can observe that our method also performs better than diffusion-based methods.
>
> | Dataset | SMD | MSL | SWaT |
> | --- | --- | --- | --- |
> | Metric | Aff-P/Aff-R/Aff-F1 | Aff-P/Aff-R/Aff-F1 | Aff-P/Aff-R/Aff-F1 |
> | ImDiffusion | 74.90/92.13/82.63 | 66.12/93.86/76.90 | 60.40/95.96/74.13 |
> | D3R | 64.87/97.93/78.02 | 66.85/90.83/77.02 | 60.14/97.57/74.39 |
> |TAD-UP|77.26/94.53/**85.03**|68.81/92.37/**78.87**|62.51/94.87/**75.36**|

---

> ### Comment · Reviewer_NV9B · 2025-11-28
>
> I genuinely appreciate the effort the authors have invested in clarifying the motivations, model design, and theoretical analysis. The responses have addressed several of my concerns. I also acknowledge that the idea of interactive learning between continuous and discrete variables is novel and indeed has the potential to enhance multivariate time-series anomaly detection.
>
> However, I still have a few remaining questions and concerns:
>
> 1. **On the covariance terms in Eq. (9) and Eq. (13).**
>    You mentioned that $L_c$ in Eq. (13) includes the covariance whereas $L_{MIE}$ in Eq. (9) does not. However, both Eq. (9) and Eq. (13) involve $\Sigma'_T$. If $\Sigma'_T$ is what you refer to as the covariance, then it appears in both losses. If the covariance terms in Eq. (9) and Eq. (13) refer to different quantities, would it be possible to use different notations to avoid ambiguity?
>
> 2. **On the added theoretical analysis.**
>    Thank you for providing additional theoretical discussion. While adding more conditions may certainly strengthen anomaly discrimination, the resulting conclusion appears relatively intuitive.
>
> 3. **On the importance of marginal probabilities.**
>    As far as I understand, the covariance term primarily captures correlations among variables. Could you further clarify how introducing covariance specifically helps the model learn more accurate *marginal* probabilities for each variable?
>
> 4. **On the 1:1 weighting between $L_c$ and $L_d$.**
>    In the final training objective, the weights for $L_c$ and $L_d$ are fixed at a 1:1 ratio. Is this ratio universally suitable for all datasets? Would it be possible to release the experimental code in an anonymous GitHub repository, to help remove uncertainties regarding implementation details?
>
> 5. **On the limitations section.**
>    The current description of the *LIMITATIONS* section in line 495 feels somewhat general. Would it be possible to provide more concrete, method-specific limitations?
>
> 6. **On Figure 1 and correlated variables.**
>       Since the authors mention that variables 1 and 5 in Figure 1 are correlated, could you please label which curve corresponds to variable 1 and which corresponds to variable 5? Additionally, it is currently unclear how the figure demonstrates their correlation.
>
>
> Once again, I sincerely thank the authors for their hard work and for taking the time to address my comments. If the authors could further clarify the above concerns, I would be happy to raise my score.

---

> > ### Author Response · Authors · 2025-11-29
> > **Response to Reviewer NV9B**
> >
> > We would like to sincerely thank Reviewer **NV9B** for the comments to improve our paper. We have revised our paper with blue color accordingly.
> >
> > **1: The covariance terms in Eq. (9) and Eq. (13)**
> >
> >  **A:** First, we would like to clarify that our covariance term is in Eq. (11) but not Eq. (13). Second, would like to clarify that we have used different notations for the variance and covariance terms in Eq. (9) and Eq. (11) in lines 314 and 327 in the original submission. Specifically, Eq. (9) uses the notation ${\hat \Sigma} _T$ to denote the diagonal variance matrix that has no covariance terms. Eq. (11) uses the notation ${\Sigma} _T$ to denote as the empirical covariance matrix that has covariance terms. The decoder cannot output the covariance matrix that should be positive-definite. Thus, we have two different terms and notations in Eq. (9) and Eq. (11).
> >
> > **2: Theoretical analysis**
> >
> >  **A:** We would like to clarify that our theoretical analysis is not just intuitive. In lines 809 and 832 of the revised paper, we provide two quantitative theoretical equations to analyse the conditional influence between continuous and discrete ODEs. In line 861 of the revised paper, we provide another quantitative theoretical equation to analyse the ratio of volumes for boundary of anomaly detection. All these quantitative equations provide that our theoretical analysis is not just intuitive.
> >
> >
> > **3: Covariance and more accurate marginal probabilities**
> >
> >  **A:** In a Gaussian Mixture Model (GMM), allowing a full covariance matrix significantly improves the accuracy of the marginal probability estimates for each variate. This happens for two main reasons: (1) the diagonal elements of the matrix $\Sigma$ determine the marginal variances and probabilities directly, and (2) the off-diagonal covariance terms control the variances estimation.
> >
> > For a multivariate Gaussian distribution $p ( \mathbf{x} )  =  \mathcal{N} ( \mathbf{x} \mid \mu, \Sigma),$ the marginal distribution of variate $x_i$ is $p(x_i) = \mathcal{N}(x_i \mid \mu_i,\, \Sigma _{ii}),$ which shows that the marginal variance is exactly the $i$-th diagonal entry of $\Sigma$. If the covariance matrix is restricted by diagonal matrix, then all marginals are forced to learn independent variances, leading to inaccurate variances estimation. Thus, introducing covariance matrix with off-diagonal covariance terms will help learn more accurate marginal probabilities.
> >
> >
> > **4: Weighting between $\mathcal{L} _ {c}$ and $\mathcal{L} _ {d}$, reproducibility **
> >
> >  **A:** First, we would like to clarify that we have used 1:1 weighting between $\mathcal{L} _ {c}$ and $\mathcal{L} _ {d}$ as shown in line 342 in our original submission. We do not introduce more hyperparameters for weighting between $\mathcal{L} _ {c}$ and $\mathcal{L} _ {d}$ as we use Maximum Likelihood Estimation in the probabilistic space instead of variate-value space. Thus, our method is capable of state-of-the-art accuracy across nine datasets without introducing more hyperparameters for weighting between $\mathcal{L} _ {c}$ and $\mathcal{L} _ {d}$.
> >
> > Second, we would like to clarify that we have made every effort to ensure reproducibility as shown in line 486 in our original submission. Our code will also be publicly available upon acceptance.
> >
> > **5: Limitations section**
> >
> >  **A:** We have revised our paper in line 495 to provide more method-specific limitations as suggested: there may be variates that show both continuous and discrete properties, such as the number of service connections. These variates change in $\mathbb{N}$ as discrete dynamics, and their slopes can also be calculated as continuous dynamic. Even existing open benchmark datasets do not include such variates, it is of interesting to model them.
> >
> > **6: Figure 1 and correlated variates**
> >
> >  **A:** We have revised our paper in line 84 according to the Figure 1. Specifically, variate 1 denotes the latency and variate 5 denotes the system flag, as shown in the Figure 1. When the system has different latencies across time, the system flag is correlated to change. Such correlation is also shown in the Figure 1.

---

### Official Review · Reviewer_1riv · 2025-10-27

**Soundness:** 3
**Presentation:** 3
**Contribution:** 3
**Rating:** 8
**Confidence:** 3

**Summary:**

The paper presents a method that learns both continuous and discrete dynamics for time-series anomaly detection and uses these dynamics to estimate correlations among continuous and discrete variables. Results across nine benchmarks are promising.

**Strengths:**

This is a well-written document, and the proposed scheme appears sound. The authors report results on nine benchmarks and compare against 21 baselines.

The paper employs Neural Ordinary Differential Equations (NODEs) to learn temporal dynamics. Prior work (e.g., Jin et al., 2023) has shown that NODEs can model continuous dynamics in continuous time. This paper further demonstrates how NODEs can model dynamics for discrete events and proposes a way to learn correlations among variables.

The ablation study is well designed.

The supplementary material includes important details that help one understand the work carried out in this document.

**Weaknesses:**

It is not clear how the trained model is used at inference time. Section 3 frames the task as Time Series Anomaly Detection, but it is silent on how the learned models—which output correlations between continuous and discrete variables—are applied to detect anomalies within a given window. A brief, explicit description of the inference procedure would help.

Relatedly, does the method require an anomaly threshold? If not, why is it unnecessary? If it does, how is the threshold chosen?

The paper will benefit from a limitations sections.  Are there any?

**Questions:**

I can see how one might construct dense features at each time step t_i for continuous variables. How is this done for discrete variables? (Ln. 224)

I did not fully understand Eq. (4). How does the gating work? Is there no sum-to-one constraint? The same question applies to Eq. (7). Also, in Eq. (7), what happened to \lambda(t)?

In Section 4.5, why is it not appropriate to model correlation among discrete variables with a Gaussian distribution? Recall that discrete events are converted into dense features at each time t_i, so one would imagine that it is possible to compute correlations between discrete variables using Gaussina distribution.  And why does this correlation depend on the horizon T?

My understanding is that the losses in Section 4.5 are used to train the model. Could you expand on how the ground-truth targets were defined to compute these losses?

---

> ### Author Response · Authors · 2025-11-25
> **Response to Reviewer 1riv**
>
> We would like to sincerely thank Reviewer **1riv** for the comments to improve our paper. We have revised our paper with blue color accordingly.
>
> **W1: Inference**
>
>  **A:** We have revised our paper in line 344 as follows to make inference stage more clear: To align with existing time series anomaly detection methods, our problem definition and experimental settings are also detecting time series anomaly at each timestamp $t_i$. During inference, joint probabilities of all data observations denote the anomaly scores, i.e., a low joint probability at a timestamp $t_i$ indicates a likely anomaly. As our object function is the Log probability, we compute the object function with data observations and the predicted distributions as the anomaly scores.
>
> **W2: Threshold**
>
>  **A:** We have revised our paper in line 371 and Appendix D.2 to make the threshold strategy more clear. Following the existing settings in state-of-the-art methods such as D3R and DADA, we use the same strategy, called SPOT, to choose the threshold of anomaly scores for all methods when computing binary anomaly labels and values for the metrics Aff-Precision, Aff-Recall, and Aff-F1. The ROC-AUC metric does not require a threshold as it considers the average accuracy under all possible thresholds.
>
> **W3: Limitations Section**
>
>  **A:** We have revised our paper in line 494 with Limitations Section. In this work, we have provided theoretical analysis and experimental results of how continuous and discrete dynamics will mutually reinforce each other in multivariate time series anomaly detection. In future research, it is of interest to explore efficient hyperparameter tuning and pre-training our model on the large-scale multi-domain time series data.
>
> **Q1: Embedding layer**
>
>  **A:** We have revised our paper in line 231 to make the embedding layer for discrete variates more clear. Like discrete word embedding in Natural Language Processing, our embedding layer for discrete variates is one-hot based lookup embedding layer (implemented by Pytorch EmbeddingBag), where discrete variates are presented using natural number encoding.
>
> **Q2: Gating and $\lambda(t)$**
>
> **A:** Our gating is based on the sigmoid activation function and element-wise Hadamard product to integrate information from continuous to discrete, and vice versa, which is like the gating process in GRU. Thus, there is no sum-to-one constraint like Transformer.
>
> $\lambda(t)$ is learned from the context discrete dynamics $\mathbf{Z}(t)$, which controls the conditional intensity of the next eventual time point for the jump in discrete variates. As shown in Figure 3, the larger $\lambda(t)$ is learned, the sooner the next eventual time point will occur, e.g., the discrete variates will change in 1 second. The smaller $\lambda(t)$ is learned, the slower the next eventual time point will occur, e.g., the discrete variates will change after 10 seconds.
>
> **Q3: Gaussian distribution, discrete variates, and correlation on $T$**
>
> **A:** Although discrete events are embedded into dense features, the decoder is designed to transfer dense features back to discrete events. For example, our decoder will output that the system has possibility of 0.8 in status 1. However, there is no possibility for status 1.1 as the status is discretely numbered. Fitting a Gaussian distribution for discrete variates will output incorrect status such as status 1.1. Thus, the Softmax distribution is better suited for discrete variates.
>
> As we design the correlations depending on the time horizon $T$, there will be more model capabilities, compared to only static correlation. Because it is commonly seen that correlation strength is different at different time.
>
> **Q4: Ground-truth targets and losses**
>
> **A:** The losses in Section 4.5 are derived from the Maximum Likelihood Estimation (MLE). Take continuous variates as example, the ground-truth targets are the observed continuous data $c_T$ at each time $T$ in the sliding window. MLE assumes that the ground-truth targets are observed because it has largest possibility. $N ( \hat \mu _T, \hat \Sigma _T)$ is the predicted distribution from the decoder. With the Maximum Likelihood Estimation losses in Section 4.5, the model will learn to output correct $\hat \mu _T $ and $\hat \Sigma _T$ to make that the ground-truth targets (the observed data $c_T$) have largest possibility under the predicted distribution $N ( \hat \mu _T,\hat \Sigma _T)$.

---

### Official Review · Reviewer_iAmw · 2025-10-30

**Soundness:** 3
**Presentation:** 4
**Contribution:** 4
**Rating:** 8
**Confidence:** 3

**Summary:**

This paper proposes TAD-UP that learns both continuous and discrete dynamics for Time series Anomaly Detection via Unified Probabilistic modeling. The contributions of this paper are as follows:
1. Proposes the first method capable of learning both continuous and discrete dynamics for multivariate time series anomaly detection.
2. Proposes gated co-dependent NODEs with compound Poisson process to learn correlated continuous and discrete dynamics.
3. Models joint probability distribution across different dynamics and optimize the model with MLE.
Results show that TAD-UP is capable of state-of-the-art accuracy on multiple real-world datasets and better efficiency tradeoff.

**Strengths:**

Originality and Significance: This is the first work trying to learn both continuous and discrete dynamics for multivariate time series anomaly detection. The proposed gated co-dependent NODEs with compound Poisson process is novel for fusion continuous and discrete variates. This modeling approach enables the demonstrated state-of-the-art performance, providing a more accurate and efficient tool for multivariate time series anomaly detection.
Quality: Overall the solution is clearly motivated and reasonably implemented, and corresponding evaluations are comprehensive.
Clarity: The paper is well written and easy to follow.

**Weaknesses:**

1. Although the proposed method combining continuous and discrete dynamics is intuitive and yields convincing experimental results, it appears to lack sufficient theoretical analysis or further experimental interpretation of how continuous and discrete dynamics mutually reinforce each other in multivariate time series anomaly detection.
2. For datasets containing only continuous variables, such as Creditcard, there is no explanation of how each module in the model are simplified to handle only continuous variables. Please clarify.

**Questions:**

1. I wonder if it's possible to provide some visualizations of ablation results and analyses to explain how continuous and discrete dynamics mutually enhance multivariate time series anomaly detection.
2. Taking the continuous co-ODE as an example, I understand that the author employs a gated TCN approach to introduce the discrete dynamics Z(0) during the process of dH(t)/dt. However, could this notation be optimized to better represent Z(0)'s involvement? For instance, dH(t)/dt|Z(0). Since this aspect constitutes the core contribution of the paper, highlighting it in the notation would be preferable. This suggestion also includes the representation of ODESolver(.) in Figure 2.

---

> ### Author Response · Authors · 2025-11-25
> **Response to Reviewer iAmw**
>
> We would like to sincerely thank Reviewer **iAmw** for the comments to improve our paper. We have revised our paper with blue color accordingly.
>
> **W1 and Q1: Mutually reinforce**
>
>  **A:** We have revised our paper in Appendix C and E.1, including theoretical analysis and further visualizations.
>
> **First**, our method can model the conditional continuous and discrete dynamics $\frac{dH(t)}{dt}\Big| _{Z(0)}$ and $\frac{dZ(t)}{dt}\Big| _{H(0)}$. More theoretical analysis is provided in Appendix C.1.
>
> **Second**, the mutual reinforcement of our conditional continuous and discrete dynamics is resulted from the mutual information: a higher mutual information concentrates the joint density of normal samples and therefore makes anomalous samples more separable (this theoretical analysis is provided in Appendix C.2 and below).
>
> The mutual information of $C$ and $D$, where $C$ denotes continuous variates
> and $D$ denotes discrete variates, is given by:
> $I(C;D)=KL\big(p(c,d)\big\|p(c)p(d)\big)= E(C) - E(C\mid D)= E(D) - E(D\mid C),$
> where $E(\cdot)$ denotes entropy. As usually holds that continuous and discrete variates are correlated, we have $I(C;D) > 0$, $E(C) > E(C\mid D)$, and $E(D) > E(D\mid C)$.
>
> For continuous variates $C$ with density $p(c)$, the traditional time series anomaly detection methods assume that normal samples lie within a small set $\mathcal{T}_C$ that has large probability density whose volume satisfies
> $ \mathrm{Vol} ( \mathcal{T}_C ) \approx \exp\big(E(C)\big).$
>
> When our method models the conditional dynamics, the conditional distributions
> $p(c\mid d)$ and $p(d\mid c)$ will be sharper than $p(c)$ and $p(d)$, as $E(C) > E(C\mid D)$, and $E(D) > E(D\mid C)$. Thus, conditioning on $D$ yields a smaller set $ \mathcal{T} _ {C \mid D}$  that has larger probability density:
> $ \mathrm{Vol} ( \mathcal{T} _ {C \mid D} ) \approx
> \exp\big( H ( C \mid D ) \big),$
> and the ratio of volumes satisfies
> $\frac{\mathrm{Vol}(\mathcal{T}_{C\mid D})}{\mathrm{Vol}(\mathcal{T}_C)}
> =\exp\!\left(H(C\mid D)-H(C)\right)=\exp\!\left(-I(C;D)\right).$
>
> The decision boundary between normal and abnormal data has been improved.
> Thus, the mutual reinforcement between the continuous and discrete dynamics enhances anomaly detection.
>
> **Third**, we provide further visualizations in Appendix E.1. From Figure 4(a), we can observe that continuous and discrete variates are correlated. From Figure 4(b), we can observe that the discrete variates are anomalies conditional on other variates. From Figures 4(c) and (d), we can observe that the decision boundary between normal and abnormal data has been improved with our method, which is consistent with our theoretical analysis above.
>
> **W2: Only continuous variables**
>
> **A:** We have revised our paper accordingly in Appendix D.1. For datasets containing only continuous variables we simply remove the corresponding discrete co-ODE branch, $\operatorname{TCN_{gate_{2}}}$, and cross entropy branch.
>
> **Q2: Notation**
>
> **A:** We have revised the Figure, Section 4 and Appendix C accordingly, using the suggested notation to highlight our conditional continuous and discrete dynamics.

---

### Author Response · Authors · 2025-11-29

Dear Area Chairs and Reviewers,

We would like to sincerely thank you for your dedication to our submission. Here we would like to summarize and emphasize some key aspects:

**Our contribution**: We propose TAD-UP that learns interactive continuous and discrete dynamics for time series anomaly detection via unified probabilistic modeling and novel gated co-dependent NODEs. TAD-UP is capable of state-of-the-art accuracy on nine datasets.

This submission has received positive scores from two reviewers, and the novelty and contributions of our work are acknowledged by all three reviewers:

(1)	Reviewer iAmw: “first method capable of learning both continuous and discrete dynamics” “proposed gated co-dependent NODEs with compound Poisson process is novel” “demonstrated state-of-the-art performance” “clearly motivated”

(2)	Reviewer 1riv: “this paper further demonstrates how NODEs can model dynamics for discrete events” “results across nine benchmarks are promising” “ablation study is well designed”

(3)	Reviewer NV9B: “co-dependent neural ODEs is interesting” “ a new perspective for handling heterogeneous multivariate time series” “acknowledge that the idea of interactive learning between continuous and discrete variables is novel”

**Summary of rebuttal**: Following the insightful suggestions from reviewers, we further improved our paper in the following aspects:

(1)	Added theoretical analysis and visualizations: iAmw and NV9B

(2)	Clarified only continuous and conditional notation: iAmw

(3)	Clarified inference, threshold, embedding layer, gating, gaussian distribution correlation, losses and limitations: 1riv

(4)	Clarified motivation, novelty, compound Poisson process, gaussian capacity, inference and margin probability and losses: NV9B

(5)	Added more baselines: NV9B


**During rebuttal**: Reviewer NV9B (1) has presented that our first responses have addressed several concerns; (2) has acknowledged that our interactive learning between continuous and discrete variates is novel; (3) would be happy to raise the score if the authors clarify some additional concerns.

We believe that we have carefully addressed all initial and additional concerns of three Reviewers, case by case.

Hope this summary will be helpful for the ACs in reducing their workload and making the final decision.

We would like to sincerely thank you again for your dedication.

Best regards,

Authors of submission 1761

---

### Meta-Review · Area_Chair_yyN9 · 2026-01-03

**Summary:**

This paper proposes TAD-UP, a framework for multivariate time-series anomaly detection that jointly models continuous-time dynamics and discrete event dynamics via two co-dependent branches coupled through a gating mechanism, with anomalies detected using joint likelihood-based scoring. Reviewers agree that the work is generally well motivated for heterogeneous time series, and that the experimental evaluation is broad, covering nine datasets with extensive baseline comparisons and competitive performance. However, while the system is competently engineered and empirically strong, the methodological novelty and positioning with respect to existing continuous-only and discrete-only anomaly detection methods are not articulated sharply enough. In particular, it remains unclear what the primary technical advance is beyond integrating known modeling components into a unified latent and probabilistic framework, which makes it difficult to assess the contribution as a clearly differentiated methodological step forward.

**Reviewer Concerns:**

The rebuttal improved clarity and addressed a number of technical questions, but uncertainty remains: whether the paper’s contribution represents a well-justified new methodological approach or is primarily an integration of existing modeling components. While the joint modeling of continuous and discrete dynamics was viewed as interesting and empirically effective, some reviewers felt that the approach largely combines known techniques, making it unclear what the core methodological advance is beyond integration. Concerns were also raised about the theoretical analysis, which was seen as largely intuitive and not tightly connected to the specific modeling choices. Overall, despite improvements in presentation and clarification, questions remain about whether the paper sufficiently establishes a distinct and well-justified technical contribution.

**Reviewer Scores:**

The two positive reviews are likely to remain positive. The most negative review is likely to be raised following the rebuttal and clarifications. However, these potential shifts in a positive direction are not sufficient to outweigh the Area Chair’s main concern after careful consideration of the reviews and the paper.

---

### Decision · Program_Chairs · 2026-01-26

Reject